# Position: It is Time to Virtualize Foundation Models with a Self-evolving Operating System Layer

**Suparna Bhattacharya** [1]  **Tarun Kumar** [* 1]  **Cong Xu** [* 1]  **Satish Kumar Mopur** [1]  **Jiahao Li** [1]  **Ashish Mishra** [1]
**Aalap Tripathy** [1]  **Annmary Justine Koomthanam** [1]  **Martin Foltin** [1]  **Ian Foster** [2]

## Abstract

AI applications have shifted from single, monolithic foundation models (FM) to compound agentic systems. Yet today's stacks remain fragmented: even as protocols (e.g., MCP, A2A) ease tool/agent connectivity, each framework embeds an implicit runtime for state, memory, budgets, and guardrails, making behavior non-portable and governance brittle. It mirrors computing before operating systems, when every program reimplemented basic services. This position paper argues that the field now needs a Foundation Model Operating System (FMOS): a system layer that virtualizes FM interactions analogous to how virtual machines abstract physical hardware, giving applications the illusion of dedicated, trustworthy FM instances with effectively unbounded capabilities. Internally, the FMOS orchestrates knowledge across memory tiers, model selection and resource allocation, and verification and policy enforcement. Like the human brain switching between fast intuition and slow deliberation, the FMOS learns when to intervene and when to let inference proceed directly and continuously adapting its policies based on operational experience.

## 1. Introduction

The excitement surrounding Foundation models (FMs) has led to widespread adoption across industries, from healthcare and finance to manufacturing and customer service, triggering a shift to compound AI systems (Zaharia et al., 2024) where multiple models and modules cooperate (Kandogan et al., 2024; Santhanam et al., 2024; Wu et al., 2023;

Zhuge et al., 2024; Liu et al., 2025). New computational workloads are being produced in which numerous agents seek to access information, computing power, physical devices, etc., to address tasks that may be bounded ("predict tomorrow's weather") or open-ended (Hughes et al., 2024) (e.g., "learn more about catalysts") or both (e.g., collaborate to "detect, resolve, and prevent incidents in IT operations" for autonomous data centers). Harnessing FMs effectively in such workloads requires self-evolving capabilities whereby an FM system's knowledge is continuously augmented with accurate data, logic, and new observations; reasoning is progressively enhanced, adapted, and verified to ensure compliance with growing expectations; and utilization is optimized to meet resource constraints for desired use cases.

Yet today's agent stacks remain fragmented. Each framework implements its own cross-cutting services—context management, tracing, tool execution, and verification—forcing some applications to rebuild components without substrate reuse. Even context management alone varies substantially across harnesses: some maintain passive compaction, while others proactively externalize artifacts to agent filesystems (Cursor, 2026). The result is a pre-operating-system situation: core services are reimplemented repeatedly, improvements do not propagate, and system-level optimization and governance are difficult to enforce.

In this position paper, we argue that the rapid shift from standalone FMs to *compound* agentic systems has created a missing *system layer*. We contend the necessity of a new abstraction to address this gap: *virtual foundation models* (VFMs) implemented by a *foundation model operating system* (FMOS). The FMOS virtualizes access to physical FMs and exposes a stable VFM interface while providing shared, learnable services for context and memory management, knowledge augmentation, reasoning control, resource allocation, and safety/trust enforcement. By moving these concerns beneath the application, the FMOS lets developers write agent logic against a consistent abstraction, while the system layer continuously optimizes, upgrades, and governs execution across heterogeneous workflows and agentic loops—enabling compound systems to evolve at scale.

---

[*]Equal contribution  [1]Hewlett Packard Enterprise  [2]Department of Computer Science, University of Chicago & Argonne National Laboratory. Correspondence to: Suparna Bhattacharya <suparna.bhattacharya@hpe.com>.

*Proceedings of the 43rd International Conference on Machine Learning*, Seoul, South Korea. PMLR 306, 2026. Copyright 2026 by the author(s).

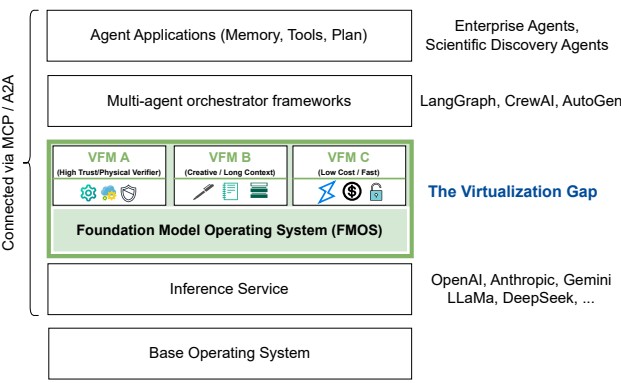

*Figure 1.* FMOS decouples agent logic from model resources, enabling scalable, secure and self-evolving agents. VFMs provide an illusion of infinite dedicated resources to applications.

## 2. Missing System Layer for Agentic Systems

Compound agentic systems are quickly becoming the dominant deployment pattern for foundation models, yet the stack lacks a *system layer* analogous to what operating systems provided for traditional software: stable execution semantics, shared services, and enforceable governance.

### 2.1. The Current Fragmentation: Proliferating Frameworks Without Common Foundations

Despite rapid progress toward compound FM systems, the supporting software stack remains fractured. In practice, adopting an orchestration framework (e.g., AutoGen, LangChain, Claude Code) means inheriting a framework-specific *runtime* that fixes execution semantics: how prompts are assembled, how state is represented, how tool outputs are retained, how failures are retried, how budgets are tracked, and how safety checks are applied. Because most semantics not exposed as portable interfaces, two logically similar agents can exhibit materially different behavior, reliability, and governance properties across harnesses.

Protocol efforts reduce friction at the **integration layer**. Model Context Protocol (MCP) and Agent-to-Agent (A2A) standardize connectivity to tools/resources and capability discovery (Anthropic, 2025b; Google Developers, 2025). But they do not define the **system layer**: portable contracts for reliable, governable execution. These gaps remain:

- **State and memory semantics**: cross-session identity, persistence, sharing, and replay/checkpointing.
- **Observability and auditability**: traces with LLM requests, tool calls, provenance, and decision paths.
- **Resource governance**: budgets, quotas, and multi-tenancy across tools/models/verification.
- **Trust enforcement**: policy application (Kumar et al., 2026), escalation, and safe-by-default mediation for sensitive operations.
- **Model mediation**: routing/caching/materialization under

controlled upgrades and rollbacks.

Absent these contracts, teams rebuild a bespoke "mini-platform" inside each framework. Improvements do not propagate across applications, and system-level optimization and governance remain brittle—a pre-OS pattern where libraries existed, but shared execution semantics did not. Recent systems gesture in this direction—AIOS (Mei et al., 2024), MemGPT (Packer et al., 2023), and Llumnix (Sun et al., 2024) explore scheduling, memory virtualization, or serving-level orchestration—but the field still lacks a unifying virtualization boundary that *jointly* governs knowledge, model reasoning, verification, and trust under one stable abstraction. Our claim is that this unified boundary is the missing system layer required for compound systems to be reusable, governable, and optimizable across applications.

### 2.2. Unifying Workflows and Agentic Loops: The Missing Execution Substrate

Enterprise deployments increasingly mix two execution regimes: (i) *workflow-driven* pipelines with explicit structure and auditability (Anthropic, 2024), and (ii) *agentic loops* that plan-act-reflect over long horizons (LangGraph, 2025; Willison, 2025; Anderson, 2025; Schmid, 2025). Today, these regimes rarely share a common substrate. Workflows assume typed steps, stable boundaries, and predictable logging; loops assume open-ended control flow, opportunistic tool use, backtracking, and adaptive context growth. Frameworks encode these assumptions into incompatible runtimes and state formats, so composing workflows and loops requires fragile glue code and yields inconsistent observability and policy enforcement at precisely the seam where enterprise guarantees matter most.

As shown in Figure 1, the core gap is the absence of portable system-layer contracts for *state, memory, budgets, and mediation* that apply uniformly across execution forms: checkpoint/resume for long-running agents, artifact persistence and retrieval, hierarchical cost/latency/tool quotas, and principled escalation to verification for sensitive actions. A system layer with a single virtualization boundary can treat workflows and loops as two schedulable *execution forms* over shared primitives, enforcing the same context/memory management, tracing, resource governance, and trust controls regardless of whether the next step is "run a node" or "plan the next move." This shared substrate enables hybrid systems that combine workflow stability with loop flexibility without sacrificing reproducibility or governance.

## 3. Position: Virtual Foundation Models Enabled by an FMOS

We formalize our position: agent deployments now require a distinct system layer—a Foundation Model Operating System (FMOS)—whose primary abstraction is the *Virtual*

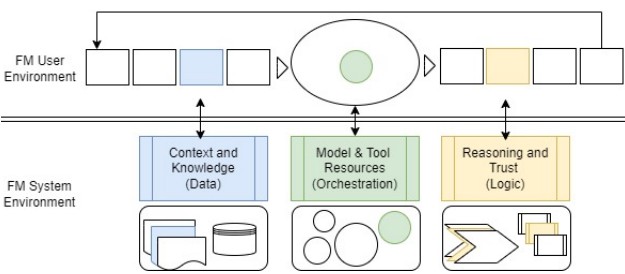

*Figure 2.* System-environment services enabling agent workflows

*Foundation Model* (VFM). A VFM presents applications with the illusion of a dedicated, trustworthy FM instance with effectively unbounded capabilities, while the FMOS mediates how knowledge is retrieved and updated, how models and reasoning are tailored to tasks, and how resources are allocated under explicit cost/latency/safety budgets.

**Self-evolution as a core principle.** Unlike traditional OS virtualization, which preserves fidelity to underlying hardware, FMOS virtualization is designed for *progressive quality gain through self-evolution*. From longitudinal interaction traces, the FMOS learns to update prompts and memories—enabling the system to improve without retraining underlying models. This evolution is managed through versioning, canary deployments, and rollback mechanisms, ensuring that improvements propagate safely across applications while maintaining reproducibility when required.

**Why Now?** Three developments make FMOS timely. First, **enterprise agentic deployments have outpaced infrastructure**: organizations are scaling beyond pilots but lack adequate governance and integration frameworks. Second, **protocol standardization has reached critical mass**: MCP and A2A enable tool/agent interoperability, but system-layer abstractions for knowledge management, reasoning verification, and policy enforcement remain missing. Third, **compound systems have proven superior to monolithic scaling**: state-of-the-art FMs are themselves compound architectures, validating that the future lies in flexible orchestration rather than ever-larger individual models. A surge of "personal AI assistant" stacks—e.g., Moltbot, Agent Zero, and Claude Cowork (Heim, 2026; AgentZero, 2026; Rogers, 2026)—signals demand for agents that operate over local files. They are natural FMOS testbeds because they require sandboxing, durable memory, and policy-governed action mediation, and would benefit from virtualization semantics.

This system-layer framing accomplishes three objectives:

1. **Unified services with co-evolution**: Context management, tool orchestration, and verification are handled at the FMOS layer; learned improvements propagate across dependent workloads.
2. **Joint optimization**: The FMOS coordinates knowledge retrieval, model selection, inference quotas, and verification depth as a combined optimization problem—

achieving efficiencies that fragmented stacks cannot.
3. **Cross-enterprise reusability**: Domain-specific skills, knowledge augmentations, and reasoning policies can be defined once and reused across applications and teams.

We formalize these three objectives in Appendix A and summarize them in A.5. Realizing FMOS requires collaborative research to define abstractions, learning mechanisms, and governance frameworks that make FM virtualization practical and principled.

## 4. Virtual FM System Environment Services

We now outline some key mechanisms and FMOS system-level services that are required to realize a virtual FM environment. First we discuss virtualization conditions and then some examples of system environment services.

### 4.1. FM Virtualization Conditions

Just as virtual memory allows applications to behave as if they have access to (potentially) unbounded memory, a virtual foundation model provides AI applications with the ability to provision and request foundation models with (potentially) unbounded capabilities.

We draw parallels from virtualization requirements in computer architecture (Popek & Goldberg, 1974) (details in Appendix A). A virtual environment (virtual machine) provided by a virtual machine monitor (VMM), is characterized by three key properties: efficiency, resource control (safety) and fidelity (equivalence). An architecture is virtualizable if the set of sensitive operations is a subset of privileged operations, where non-privileged operations execute natively while privileged operations trap if invoked from user environment, thus passing control to the VMM.

Analogously, VFMs virtualize higher-level FM capabilities such as knowledge, models, and reasoning. The key properties of efficiency and resource control are applicable in terms of these resources (e.g., context windows or reasoning operations). The third property is not fidelity, but rather progressive quality gain through self-evolution.

LLMs/FMs have been informally likened to processors that interpret human language. Thus the notion of what constitutes sensitive and privileged operations is far more complex to specify (than it is for fixed ISA hardware processors) and needs to be *learned* (offline or in-context) based on the nature of capabilities (knowledge, logic) controlled by VFMs, and potentially customized using a model virtualization protocol for FM traps, similar to how MCP enables tool calling for FMs trained with function calling capability. Once a FM trap is initiated (e.g. a knowledge trap) the FMOS capabilities activated (such as knowledge augmentation) also need to be learnable so they can evolve over time.

## 4.2. VFM System Environment Requirements

A VFM must support a prototypical FM user workflow (Figure 2) comprising input context preparation, model execution passes, and output processing—in a manner that allows for transparent system-level interception and control.

Such interception enables system-level (FMOS) services that constitute the underlying FM system environment, which supports underlying capabilities for simplifying and optimizing FM-based agent applications and for enabling the system's capacity for self-evolution.

Self-evolution often involves closed-loop, trajectory-driven adaptation and may use both parametric and non-parametric adaptation. Rather than necessarily fine-tuning weights or just adding data, FMOS learns from interaction traces and updates prompts, policies, and structured memories for the VFM. The system improves behavior through curated, FMOS-managed learning—enabling standardized interception, diagnosis, and safe rollout (versioning, canaries, rollback) across models and applications.

## 4.3. Context Management & Knowledge Augmentation

A FM combines internal (parametric) knowledge acquired during training with (non-parametric) knowledge that it receives as input context (prompts). System environment services control this context both to elicit (selectively focus on) what the model knows and to expand (augment) it with external knowledge. Appendix C describes a few capabilities that fall under this category, such as (1) context memory management, (2) knowledge compression and retrieval, and (3) handling knowledge-oriented abstractions for different data modalities. A key challenge in realizing these services is learning to adapt to what is most relevant for the FM application and current context.

**The gap: context-management policies as first-class objects.** Today's agent stacks lack a portable way to bind an application's *intent* for context (what must stay in-window, what can be summarized, what must be recoverable) to the *mechanisms* that actually construct prompts and manage tool outputs. As harness-specific defaults silently determine behavior, default behaviors already diverge: Claude Code compacts long histories by summarizing key decisions and continuing with recently accessed files (Anthropic, 2025a); Cursor externalizes long tool outputs (and even chat history) into files that the agent can re-read on demand (Cursor, 2026); OpenCode auto-compacts near the context limit and resumes from a summary (opencode-ai, 2026).

This missing interface matters because application developers often *know* which pieces of context are valuable (and when), but cannot express that knowledge to the serving layer. As described in Appendix C, developers may want to specify rules such as: (1) post-answer offload (appro-

priate, e.g., for Web search agent), (2) tool-output offload (applicable, e.g., for Enterprise infrastructure agent), (3) adaptive agent skills unloading, (4) retain thoughts, prune observations (appropriate, e.g., for Deep research agents).

These examples share a common structure: each is an application-level *policy* over a system-level *mechanism* (buffering, summarization, pruning, offloading, retrieval). The absence of a policy-to-mechanism mapping forces developers to either (i) accept brittle defaults embedded in a particular harness, or (ii) reimplement context plumbing in application code, undermining composability and reuse. This is precisely the kind of cross-cutting concern that operating systems absorbed historically: applications should express *intent* (what must be retained, what can be externalized, what must be recoverable), while the system layer enforces it efficiently under changing resource constraints.

Declarative context-policy interface enables VFMs to expose stable semantics while allowing the FMOS to learn & optimize the concrete realization of those policies over time.

## 4.4. Reasoning and Trust Augmentation

Reasoning is essential both for discovering or evolving (and integrating) new knowledge and when leveraging existing knowledge, tools, simulators, etc., and for ensuring safe, trustworthy FM outputs. System environment services can control FM output selection and processing (e.g., through constrained decoding, sampling, invoking verification and planning tools, representation engineering (Zou et al., 2023; 2024)). As described in Appendix C, such capabilities include (1) expanding and managing reasoning resources, (2) switching between reasoning at multiple tiers such as abstract and specialized reasoning, and (3) low-overhead verification, protection, and steering mechanisms.

## 4.5. Model Resource Sharing and Orchestration

FM inference demands significant GPU resources, which escalate with inference-time scaling and multi-agent setups. Distilled models can mitigate these issues. The underlying model serving platforms typically perform optimizations for all requests to a given FM, but system environment services can intercept them (Abhyankar et al., 2024) and use their awareness of higher-level intent to enable deeper co-optimizations and to manage tradeoffs involved in both model selection and orchestration. Related to the model's orchestration layer, recent work (He et al., 2025b) demonstrates that the FM serving layer can support self-evolution through concurrent execution of fine-tuning and inference. Appendix C describes three capabilities under this category: (1) Scheduling and mapping, (2) model composition and instantiation, and (3) profiling, measurement, and tracing.

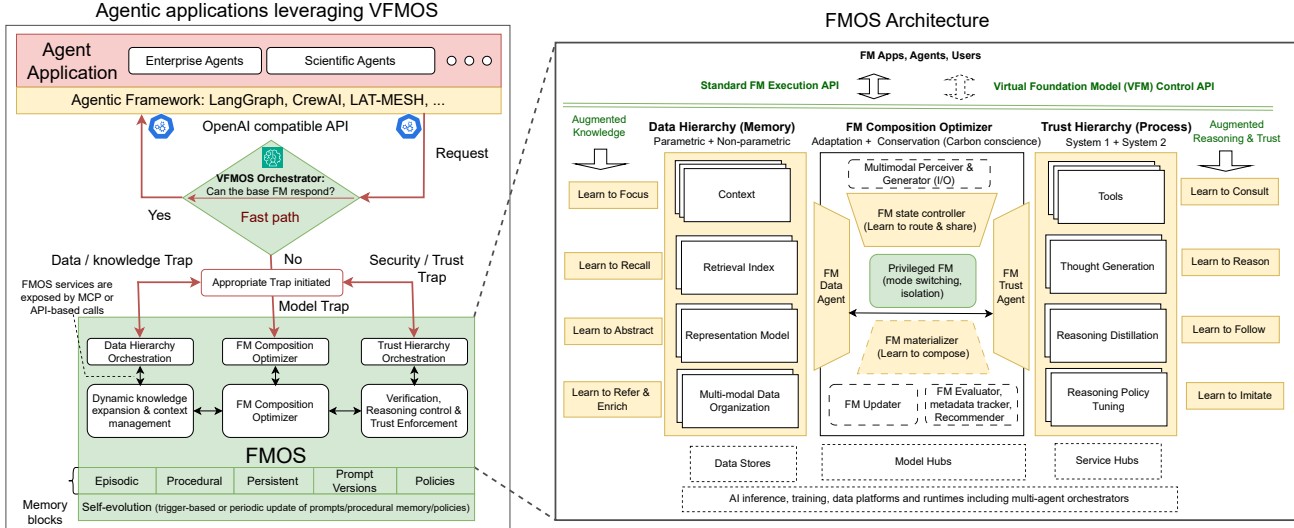

*Figure 3.* (left) FMOS intercepts the execution flow where needed and orchestrates an optimized execution path. (right) An expanded view of common fundamental capabilities offered in FMOS.

## 4.6. Broader Considerations

Beyond application and system considerations related to the three elements and their interactions, the FMOS design also involves some long term considerations:

**Continual self-evolution**: The ability to evolve and adapt is especially important for system environments that support FMs in scientific discovery and other open ended domains, where new observations, new scenarios to learn from keep emerging and new knowledge is being constantly being generated, verified and refined.

Recent studies show that the current FMs are not uniformly capable of self-evolution. Evo-Memory (Wei et al., 2025) evaluates LLM agents on self-evolving memory across streaming task settings, finds that most models fail to reliably accumulate and reuse experience across tasks. Similarly, Evo-Test (He et al., 2025a) finds that existing adaptation methods including reflection, memory augmentation, and reinforcement learning all struggle on test-time learning benchmarks. FMOS is designed to scaffold self-evolution at the system level rather than rely on the model to perform it alone. The model provides signals—uncertainty indicators, failure patterns, quality scores - while the FMOS control layer makes the actual evolution decisions: updating prompts, adjusting policies, modifying memory tiers, or rolling back unsafe changes. Model capabilities are, however, improving in this direction: recent model releases, such as MiniMax M2.7 (MiniMax, 2026), explicitly incorporate self-evolution as a native model capability, demonstrating progress in analyzing failure trajectories, planning scaffold modifications, and executing iterative optimization loops. FMOS is designed to progressively leverage such improvements as they become available.

**Continual adoption of latest techniques**: Systems environments that support emerging FM workloads must be able to continually adopt better models, frameworks and methods to keep pace with rapid advances in the AI world, so that every workflow automatically benefits from those advancements.

**External control**: System environments should have mechanisms for external controls to be applied automatically by administrators to allow incorporation of global policies, particularly with respect to safety, compliance and resource bounds, e.g. using techniques for incorporation of privileged instruction hierarchy in FMs (Wallace et al., 2024).

## 5. FMOS Architecture

Drawing on OS virtualization, we present the *VFM* in Figure 3 as the stable interface exposed to applications, while the *FMOS* mediates access to underlying *physical FMs (pFMs)* and their associated resources. By default, requests execute on a lightweight *fast path*; when task conditions warrant (e.g., insufficient context, elevated risk, or tight budgets), FMOS triggers learned *traps* to activate a more deliberative *slow path* that performs targeted knowledge augmentation, model routing/composition, and verification. Because this virtualization boundary sits beneath diverse agent frameworks and APIs, it preserves programming flexibility while enabling shared system policies and learned artifacts to evolve under standard controls (e.g., versioning, canaries, rollback). This design emphasizes three coupled challenges a virtualized VFM layer must address:

**Data and Knowledge.** Represent, expand, and continually update multimodal knowledge while managing context/ memory under finite budgets.

**Trust and Reasoning.** Adapt reasoning depth, domain-

specific verification and policy enforcement, user-specified risk levels, and time horizons.

**Efficiency and Adaptability.** Share and switch among a growing pool of models and tools to optimize quality–latency–cost trade-offs under multi-tenant constraints.

FMOS operationalizes these goals through three cooperating subsystems that sit behind the VFM interface: a *Data Agent* that manages context and non-parametric memory; a *Composition Optimizer* that selects, routes, and (when useful) composes pFMs; and a *Trust & Reasoning Agent* that governs escalation, verification, and guardrails. These subsystems are invoked through interception points (e.g., virtual model endpoints, MCP services, and framework hooks) and remain optional: a VFM can be realized as a near-direct call to a base model, or as a progressively richer orchestration that optimizes multiple objectives. FMOS is grounded in three core design principles:

**1. Virtualize capabilities behind stable semantics.** expose a stable behavioral contract (state/memory persistence, budget composition, and trust escalation) so applications rely on invariants rather than implementation details. FMOS may evolve mechanisms (routing, context tiering, retrieval/-compression, verification) as long as this contract holds.

**2. Demand-driven orchestration:** keep the common case fast, and escalate only via explicit traps when quality, budget, or trust requirements require it.

**3. Ecosystem compatibility:** integrate with existing APIs, agent frameworks, and protocols so improvements propagate without forcing application rewrites.

### 5.1. Data Agent: Dynamic Knowledge Expansion and Context Management

The Data Agent implements the VFM's *knowledge plane:* it mediates what enters the model's active context and what is persisted externally, enabling reusable knowledge augmentation under explicit context and cost budgets (Section 4). Concretely, it selects, transforms, and retrieves multimodal artifacts (e.g., documents, tables, code, images, and time series) so that model execution remains grounded while state and evidence remain recoverable across steps and sessions.

Like virtual memory, it maintains a *memory hierarchy* organized by functional role rather than raw latency: a small, fast "working set" (prompt context) backed by lower tier (files, vector or graph databases) that store episodic, procedural, and semantic artifacts. Paging and caching are *content-aware:* retrieved items may be summarized, schema-fied, embedded, or replaced with retrieval handles, preserving access semantics while keeping the active window within budget. This hierarchy is driven by learnable policies:

**Learn to focus:** produce task-conditioned context slices and grounding (e.g., self-RAG (Asai et al., 2023) or adaptive zooming (Zheng et al., 2025)) to steer generation, filter irrelevant material, and reduce cost; when needed, iterate between generation, augmentation, and refinement.

**Learn to recall:** index and retrieve previously encountered artifacts via suitable representations (embeddings, graphs, files, or representation-engineered features (Zou et al., 2023; Bartoszcze et al., 2025; Hassan et al., 2025)) to support reuse without repeatedly re-deriving the same evidence.

**Learn to abstract:** when retrieval alone is insufficient, train or finetune specific components (e.g., domain embedding models, parameter-efficient adapters, distilled auxiliaries) to improve representations and extend coverage to additional modalities such as time series (Liang et al., 2024).

**Learn to refer and enrich:** pre-process, synthesize, and align heterogeneous sources so they become retrieval- and adaptation-ready; operate efficiently and, where necessary, approximately and on-demand at multiple granularities.

By virtualizing the movement of knowledge between short-term context and long-term stores, the Data Agent gives VFMs the practical illusion of boundless, continually improving access to essential information.

### 5.2. FM Composition Optimizer

The FM Composition Optimizer maps VFM calls to *physical* model resources under explicit quality–latency–cost–policy constraints. It supports both (i) *static provisioning* of a fit-for-purpose model bundle for a class of workloads, and (ii) *dynamic scheduling* at runtime (routing, caching, sharing) as task demands and resource conditions change. Decisions are informed by continual measurement and task-specific performance traces (Saranathan et al., 2025) rather than fixed, framework-level heuristics.

**FM materializer (Learn to Compose):** provisions the execution substrate for a VFM by selecting and, when beneficial, combining/merging, adapting, distilling, or editing models from a pool of candidates to meet capability and deployment constraints.

**FM controller (Learn to Route):** performs runtime model mapping/routing (Kumar et al., 2025), caching, and sharing across instantiated models, while enforcing instruction-privilege and security boundaries and respecting SLOs (latency, throughput, and cost).

**FM updater:** manages controlled evolution of materialized models via continual (un)learning and editing, with versioning and rollback consistent with FMOS governance.

**System-level training objectives:** FMOS implies a shift from purely model-level training toward system-level objectives. The FMOS control plane generates operational sig-

nals: trap activation traces (context, confidence scores, activations), routing decision outcomes, memory update events, etc. These signals enable two complementary training regimes. First, at the control-plane level, FMOS learns policies over privileged operations (routing, retrieval, verification, memory updates) from execution traces and outcome-linked feedback across applications, operating transparently to application logic. Second, upstream model training: using FM-level signals (confidence, activation patterns) and FMOS operational traces, FMOS can identify capability gaps—situations where no available physical FM meets the VFM contract—and flag these as targets for fine-tuning, continual pretraining, distillation, or model replacement.

Operationally, this subsystem plays the role of a scheduler-plus-hypervisor for model capability: it decides *what* physical models a VFM is backed by and *when* to switch, reuse, or refresh them as workloads and the model ecosystem evolve.

### 5.3. Trust & Reasoning Hierarchy

The FMOS Trust & Reasoning Agent mediates how a VFM allocates deliberation and verification under explicit safety, cost, and latency budgets. In routine cases it stays on a lightweight "fast" path; when tasks become ambiguous, high-stakes, or policy-sensitive, it escalates to slower reasoning, tool-assisted checks, and stricter guardrails, and logs the outcomes to improve future decisions.

**Learn to consult:** Decide when to invoke tools (e.g., simulators, search, checkers, human-in-the-loop) and how to integrate their outputs as evidence rather than uncontrolled context expansion. This includes selecting verifiers appropriate to the claim type and risk profile.

**Learn to reason:** Control the reasoning *cost* and *style* (planning, decomposition, reflection), including switching between fast heuristics and deliberate search. When slow-path reasoning is validated, distill reusable reasoning templates or policies to reduce future compute for similar cases.

**Learn to follow:** When constraints and domain logic are stable and recurring, update prompts, policies, or lightweight adapters so common rules and responsible behaviors are enforced without long in-context chains or repeated retrieval.

**Learn to imitate:** When gaps reflect missing coverage in the underlying models, trigger broader upgrades (e.g., continual pretraining, model replacement, or specialized reasoning modules), gated by evaluation to manage regression and catastrophic forgetting.

This hierarchy mirrors OS protection mechanisms: as actions become more privileged or risky, they trigger progressively stronger validation, scaling computational commitment with task criticality and trust requirements.

### 5.4. Minimal Overhead with Maximum Capability

FMOS is designed around a fast path. By default, requests pass through the VFM interface with minimal interception beyond lightweight accounting and tracing. When learned "traps" fire (e.g., uncertainty, policy sensitivity, budget pressure, or anomaly signals), FMOS activates only the required subset of capabilities—escalating context retrieval, switching models, or deepening verification—and then returns execution to the fast path. This preserves low latency and cost while allowing the same VFM endpoint to provide stronger guarantees when needed, and keeps context management, routing, and trust policies transparent to applications.

### 5.5. Observability and Debuggability

As all FM interactions pass through the VFM API, FMOS can emit a causal trace per invocation covering the full execution path: trap activations, context management decisions, model routing choices, and verification outcomes. This enables layered attribution—when an agent produces an incorrect answer, developers can localize the fault to application logic, context management, routing, or verification without full interpretability of the learned policy. FMOS supports checkpoint and replay: execution state at trap points (model version, memory snapshot, policy parameters) can be recorded for controlled deterministic replay.

The core observability components are: (1) typed tracepoints at each trap emitting provenance-rich events; (2) an immutable artifact registry versioning learned artifacts (prompts, routing rules, memory schemas); (3) checkpoint/replay with a frozen mode for controlled debugging; and (4) semantic debugging hooks localizing failures across application logic, context management, routing, and verification. Key evaluation metrics include trace completeness, replay fidelity, and time-to-root-cause. Much of FMOS's self-evolution involves non-parametric updates, which are substantially easier to audit and roll back; broader parametric updates are gated by canary evaluation (Section 3).

## 6. Case Studies

We highlight two agentic applications where FMOS provides key abstractions for handling complexity and enabling system evolution.

### 6.1. Acceleration of scientific discovery

Multi-agent systems increasingly support hypothesis generation, experiment planning and control, simulation, and analysis. These workloads stress three FMOS capabilities:

**Knowledge augmentation:** Scientific evidence is distributed across heterogeneous, multimodal sources (tables, figures, time series). FMOS's Data Agent provides a policy-

driven substrate that unifies visual and textual evidence via guided retrieval. For example, a query such as "Assess catalytic activity for hydrogen evolution of this $MoS_2$ microscopy tile" can trigger a knowledge trap that retrieves relevant image regions and supporting literature, retaining only task-critical context in the active window.

**Domain-aware model selection:** Scientific tasks often require both domain knowledge and strong visual reasoning. The FM Composition Optimizer routes the request to an appropriate physical FM (or a composite) based on prior task performance under compute and latency budgets.

**Verification under physical constraints:** Outputs must respect scientific priors and physical laws. FMOS escalates to a verification path when needed, invoking multi-step checks and constraint-aware reasoning. Over time, it can reuse validated associations (e.g., between defect signatures, free-energy diagrams, and polarization curves) to guide subsequent retrieval and reduce unnecessary re-computation.

### 6.2. Technical Support: Adaptive Context Management

Enterprise technical support agents use complex decision trees to troubleshoot specific customer technical issues. The challenge is to enable agents to flexibly manage both a detailed, "zoomed-in" view of the current decision-tree node and a broader, "zoomed-out" context of the overall troubleshooting path. This ensures continuity across complex support flows. Without infrastructure to fluidly switch and validate these contexts, agents risk losing track of node states, leading to guidance errors. FMOS enables this through dynamic context handling. Rather than using a fixed window, it employs a hierarchical decision tree of past interactions that evolves over time. The Data Agent routes context based on the query's position in this tree, maintaining state across sessions. This demand-paging-like approach draws from OS memory principles and avoids manual memory engineering.

## 7. Alternative Views

*FMs will become so good at everything that we will no longer need to augment them.* FMs are expanding across modalities, context length, and large reasoning models (LRMs) (Besta et al., 2025; Xu et al., 2025a), solving hard problems through inference-time scaling. It introduces new challenges including reasoning cost, "overthinking" (Appendix B), and trustworthiness (Hylak & Latent Space, 2025). Even as FMs/LRMs improve, longer inference-time reasoning traces alone cannot gather new evidence or adapt behavior in dynamic tasks, so "thinking more" is insufficient without interaction (Shen et al., 2025). Recent agentic-reasoning work instead treats capability as a plan–act–learn loop with tools, feedback, and memory—so augmentation

remains fundamental rather than optional (Wei et al., 2026). More details are given in Appendix D.

*Agent frameworks like LangChain, AutoGen will encompass everything, when combined with query and pipeline optimization techniques for compound AI systems.* Agent frameworks help with wiring, but they do not provide system-layer guarantees. Empirical evidence shows multi-agent workflows still break on validation, context loss, rollback, and coordination, yielding inconsistent state and poor recovery (Chang & Geng, 2025); developer data likewise highlights orchestration and reliability as persistent bottlenecks (Asgari et al., 2026). Optimizers inherit these gaps, and even single agents require OS-like memory virtualization to escape fixed context limits (Li et al., 2026; Packer et al., 2024). Hence an FMOS-like layer is needed for portable semantics over state, memory, and trust.

*Model Context Protocol (MCP) and Agent-to-Agent communication protocol advancements address most challenges.* By standardizing how agents *connect*—to tools, resources, and one another—these protocols provide the interoperability needed for rapid ecosystem growth. However, they intentionally stop short of specifying *execution semantics* and *governance guarantees* (Kumar et al., 2026). As deployments scale, teams still must define (and today, reimplement) the system-layer contracts of Section 2.1. Without a shared substrate, those capabilities are bolted onto frameworks or MCP servers in incompatible ways, yielding protocol-compliant but brittle "bloat" and fragmented control.

*The OS and virtualization analogy is misleading as it is a higher level layer and does not directly manage hardware resources.* Traditional OSs virtualize hardware resources while remaining largely unaware of workload intent due to separation-of-concerns principles. For agentic systems, this semantic gap has widened: workloads are expressed in terms of FM instructions, knowledge, and reasoning, where conventional OS abstractions offer limited control for efficiency, safety, and trust (Mei et al., 2024; Zhang et al., 2024). FMOS addresses this widening gap by virtualizing higher-level FM operations above the base OS, while still leveraging OS signals and mechanisms to manage environments using OS-inspired principles (Packer et al., 2023; Mei et al., 2024).

*Rather than an OS, could FMOS play a role closer to a database system?* One could argue that the right foundation for FM system management is not an operating system but a database: transactional semantics, rich query interfaces, and mature governance frameworks already address many of the consistency and auditability concerns we identify. DB-OS research (Cafarella et al., 2020; Skiadopoulos et al., 2021; Li et al., 2025a) explored precisely the inversion of the conventional relationship, proposing that an OS be built on a database rather than the other way around. There are

genuine strengths on the database side; however, we believe that the OS framing more precisely captures what FMOS does. A database, however capable, is an external service: an application must explicitly choose to call it, query it, and interpret its results. FMOS, by contrast, is designed to automatically determine when to intervene and when not to, operating transparently beneath the model interface. The interface applications interact with today is a model interface, involving generation, reasoning, tool invocation, and context management—none of which maps naturally onto query semantics. The OS analogy is specifically about creating a virtualization layer that constructs an illusion between what the application sees as the model interface and what is actually executing underneath.

## 8. Call to Action: From Position to Practice

Realizing the vision of Virtual Foundation Models enabled by an FMOS will require coordinated effort across research communities, platform builders, and open-source foundations (e.g., LF's Agentic AI Foundation (AAIF, 2025)).

**Define and Standardize Core FMOS Abstractions (Research Community).** We must first converge on a minimal, principled set of system-layer abstractions analogous to those of conventional operating systems. The ML and systems communities should jointly define the Virtual Foundation Model (VFM) abstraction, including lifecycle, isolation semantics, and fidelity guarantees. Inspired by classical virtualization results (Popek & Goldberg, 1974), standardized interfaces for context management, knowledge augmentation, reasoning control, and trust enforcement are essential.

**Develop Open FMOS Reference Architectures and Prototypes (Systems Builders)** To ground the abstractions in practice, we urge platform builders and researchers to develop open, modular FMOS reference implementations. Intercept FM execution via existing interfaces (e.g., OpenAI-compatible APIs, MCP endpoints, agent framework hooks) without requiring application rewrites. Such prototypes should support coexistence with popular agent frameworks.

**Establish Benchmarks for System-Level FM Virtualization (ML Evaluation Community)** Progress requires shared evaluation. We call for benchmarks that go beyond task accuracy to measure system-level properties: context efficiency and knowledge reuse, robustness under evolving policies and data, cost–quality trade-offs from dynamic routing and reasoning escalation, and reproducibility and auditability under FMOS mediation. Evaluation should span components to full systems, with metrics covering task performance, resource efficiency, developer productivity, system-level attribution, and longitudinal self-evolution.

**Align Protocols and Governance Mechanisms (Standards Bodies and Enterprises)** MCP and A2A enable interoperability at the integration layer; the next step is system-layer governance. Standards bodies and enterprises should define contracts for safety enforcement, privilege levels, and external control of FM behavior, treating FMOS-level controls as first-class governance mechanisms rather than application add-ons.

**Cultivate Cross-Disciplinary Collaboration and Long-Lived Testbeds (Community)** Sustained progress requires collaboration across ML, systems, and domain experts through long-running FMOS testbeds with persistent, evolving agents, and open repositories of reusable components (e.g., data agents, trust agents, model evaluators).

In summary, principled FM virtualization will not emerge from isolated optimizations but from a shared systems agenda grounded in abstractions, benchmarks, and open infrastructure. We encourage the community to treat FMOS as a new AI stack layer shaping how foundation models evolve, interact, and are trusted.

## 9. Conclusion

The "LLM as OS" metaphor (Karpathy, 2023) has gained popularity, but its OS-and-virtualization implications remain underexplored in mainstream ML research—despite being increasingly central to how compound agentic systems are built and governed. We argued that rising system complexity makes an explicit virtualization layer necessary: an FMOS that mediates access to physical FMs and exposes stable Virtual Foundation Models (VFMs). This boundary decouples application logic from context management, model routing, and trust enforcement, enabling coordinated optimization, portability, and auditable governance as systems evolve.

This position also sharpens the research agenda: what training and interfaces make models effective *system components* (e.g., for learned traps, mediation, and policy execution), and what skills distinguish FMOS agents from application agents? Continual self-evolution further introduces controlled nondeterminism (e.g., routing decisions that legitimately change with new evidence). The remedy is not to freeze adaptation, but to make it *operationally safe*: explicit versioning, traceable decision logs, and principled observability that preserve reproducibility and accountability.

We invite the community to treat VFMs as first-class research objects: formalize abstractions, build prototypes, and establish benchmarks that measure reliability, reuse, cost, safety, and longitudinal behavior under continual evolution.

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

# A. Why FMOS Enables Co-evolution, Co-optimization, and Reuse: A Virtualization Lens

## A.1. A minimal FM-virtualizability condition (and what it buys us)

An application interacts with the foundation-model stack through an operation set $\mathcal{A}$ (e.g., generate, retrieve, cite-check, tool-call, write/read memory, route to another FM). Following virtualization tradition, we partition operations into:

Innocuous operations ($\mathcal{A}_{\text{ino}}$) execute on the fast path and do not affect shared FMOS-managed state: examples include drafting a response from already-loaded context, local summarization, and formatting or paraphrasing content already present in the active window. Sensitive operations ($\mathcal{A}_{\text{sen}}$) change or consume shared resources—budgets, memory tiers, trust state, or model allocation—and therefore must belong to $\mathcal{A}_{\text{priv}}$ (Eq. 1).

FMOS designates a set of *privileged* operations $\mathcal{A}_{\text{priv}} \subseteq \mathcal{A}$ that must "trap" to the FMOS control plane (dispatcher/allocator/interpreters). We assume a minimal virtualizability condition:

$$\boxed{\mathcal{A}_{\text{sen}} \subseteq \mathcal{A}_{\text{priv}}} \tag{1}$$

i.e., every operation that can impact shared budgets/knowledge/trust is mediated by FMOS.

In a representative enterprise-agent workflow: innocuous operations include generating an answer from retrieved context, rephrasing a tool output, or running an in-context arithmetic check; sensitive operations include increasing retrieval depth, writing to persistent memory, routing to a different physical FM, invoking an external tool with side effects, or escalating to a verification or policy check. Critically, the partition is not a fixed, hand-authored list. The semantic criterion is fixed (($\mathcal{A}_{\text{sen}}$) = operations affecting shared FMOS state), but classification is adaptive at runtime: an operation like local summarization is innocuous by default, but becomes sensitive if it crosses a budget threshold, modifies persistent memory, or alters trust-relevant state. The mechanism governing reclassification is FMOS's learned trap policy (Sec. 5.4).

**Conservative guarantees (analogous to VM goals).** Under (1), FMOS can *target* three properties (we phrase them conservatively to avoid overclaim):

- **Efficiency (fast path):** operations in $\mathcal{A}_{\text{ino}}$ do not require orchestration and can execute with minimal FMOS involvement.

- **Resource control:** effects on shared resources/trust/knowledge occur only via trapped operations, enabling enforceable budgeting and policy checks *within the VFM interface*.

- **Interface-level equivalence:** applications program against a stable VFM interface (ABI); FMOS may change internal realizations while preserving agreed semantics (up to latency/noise/stochasticity).

Crucially, cross-cutting improvements (retrieval, verification, routing, memory updates) live in a small trapped surface, while most application logic remains unchanged on the fast path.

## A.2. 1) Co-evolution of capabilities (shared learning over privileged mechanisms)

**FMOS as a shared "control program" policy.** Let FMOS implement a parameterized policy class $\Pi$ over privileged actions:

$$a_t \sim \pi(\cdot \mid s_t), \quad a_t \in \mathcal{A}_{\text{priv}}, \quad \pi \in \Pi,$$

where $s_t$ summarizes request features (domain, risk, uncertainty, budget, user intent, etc.) and $a_t$ selects retrieval depth, verifier strength, routing choice, memory tier, or tool plan.

Assume $K$ applications/tenants produce traces $\tau \sim \mathcal{D}_k$ with loss $\ell_k(\tau; \pi)$ capturing quality/trust/cost tradeoffs. FMOS learns a single shared policy:

$$\pi^\star \in \arg\min_{\pi \in \Pi} \ J(\pi) \ \triangleq \ \sum_{k=1}^{K} w_k \, \mathbb{E}_{\tau \sim \mathcal{D}_k}\big[\ell_k(\tau; \pi)\big]. \tag{2}$$

**Why "co-evolution" is a real effect (not just reuse).** Let $\hat{J}_N(\pi)$ be the empirical objective formed from $N = \sum_k N_k$ trapped-operation samples across apps. For bounded policy complexity (finite $\Pi$ or standard capacity control), uniform

convergence yields:

$$\sup_{\pi \in \Pi} \left| J(\pi) - \hat{J}_N(\pi) \right| \leq O\left( \sqrt{\frac{\text{Comp}(\Pi)}{N}} \right), \tag{3}$$

where $\text{Comp}(\Pi)$ stands for $\log |\Pi|$ (finite case) or a capacity measure (e.g., Rademacher/VC/norm). If each application instead learns its own $\pi_k$ using only $N_k$ samples, its estimation error scales as $O(\sqrt{\text{Comp}(\Pi)/N_k})$, which is worse whenever $N \gg N_k$. Thus, improvements to privileged mechanisms (retrieval/verification/routing/memory) learn faster and generalize better when trained once at FMOS and shared.

**Single-application case (no overclaim).** Even with $K = 1$, co-evolution holds *over time*: a single application generates many trapped events across sessions/tasks/users, so $N$ grows and (3) still yields steadily improving virtualization policies. In addition, co-evolution applies *within* a single application when it contains multiple agents/subtasks that share FMOS.

**Failure mode:** Eq. 1 determines how much of the execution is actually mediated, and therefore how much benefits from shared control and co-evolution. When Eq. 1 is fully satisfied, FMOS provides the strongest system-level guarantees: all sensitive operations pass through a common boundary, enabling enforceable budgeting, shared policy learning, and coherent co-evolution across applications. When Eq 1 is only partially satisfied, the degradation is graceful. FMOS continues to govern the mediated subset, providing control, policy enforcement, and co-evolution over those trapped operations; guarantees are lost only for the leaked ones. The traces from missed mediations steer the trap mechanism (Sec 5.4) after self-evolution and the system attempts to enhance its recall in the future. Correspondingly, Eq. 2 is learned over only the mediated fraction of the privileged action space, and the co-evolution benefit is proportionally reduced but not eliminated. The practical failure mode is a regression toward the fragmented execution semantics rather than an architectural collapse. The partition of $\mathcal{A}$ into $\mathcal{A}_{\text{ino}}$ and $\mathcal{A}_{\text{sen}}$ is semantically defined but adaptively realized at runtime, not a fixed hand-authored list. The criterion is stable: $\mathcal{A}_{\text{sen}}$ consists of operations whose effects depend on or can change shared FMOS-managed state (budgets, memory tiers, trust state); $\mathcal{A}_{\text{ino}}$ are the remaining fast-path operations. Classification, however, is context-dependent. A local summarization is innocuous by default, but becomes sensitive if it crosses a budget threshold, modifies persistent memory, or alters trust-relevant state, at which point it traps to FMOS. The mechanism governing reclassification is the learned trap policy (Sec. 5.4):: uncertainty, policy sensitivity, budget pressure, and anomaly signals all serve as runtime triggers that promote a fast-path operation to the sensitive path. This design separates the *fixed semantic criterion* (what makes an operation sensitive) from the *adaptive runtime classification* (whether a given operation instance is sensitive in context), allowing the partition to remain principled while accommodating the context-dependence inherent in FM workloads.

### A.3. 2) Co-optimization of resources (global allocator + trust-aware budgets)

**Coupled budgets are the point.** Let there be $R$ shared resources: tokens, GPU time, tool-call quota, latency budget, memory writes, verifier invocations. At time $t$, application $k$ chooses privileged action $a_{k,t}$ with utility $u_k(a_{k,t})$ and consumption $c_r(a_{k,t})$. FMOS solves a global constrained optimization:

$$\max_{\{a_{k,t}\}} \sum_{k,t} u_k(a_{k,t}) \quad \text{s.t.} \quad \sum_{k,t} c_r(a_{k,t}) \leq B_r, \ \ \forall r \in \{1, \dots, R\}. \tag{4}$$

**Shadow prices yield coordinated decisions (under standard assumptions).** Introduce multipliers $\lambda_r \geq 0$ ("shadow prices") and consider the Lagrangian

$$\mathcal{L}(\{a_{k,t}\}, \lambda) = \sum_{k,t} \left( u_k(a_{k,t}) - \sum_{r=1}^{R} \lambda_r c_r(a_{k,t}) \right) + \sum_{r=1}^{R} \lambda_r B_r. \tag{5}$$

Given $\lambda$, each application selects actions locally:

$$a_{k,t}^{\star}(\lambda) \in \arg \max_{a \in \mathcal{A}_{\text{priv}}} \left( u_k(a) - \sum_{r=1}^{R} \lambda_r c_r(a) \right). \tag{6}$$

FMOS updates $\lambda$ to satisfy budgets (dual ascent), yielding a globally optimal allocation for (4) under convexity/regularity (and a principled heuristic otherwise). This is the formal meaning of *co-optimization*: a shared allocator sets system-wide prices/policies so that many local decisions collectively respect shared budgets and maximize total value.

**Trust/safety as a first-class coupled constraint (not an afterthought).**   Let $S(\{a_{k,t}\})$ denote an aggregate risk measure (e.g., expected policy violation / hallucination / unsafe tool side-effect rate). FMOS can enforce a risk budget:

$$\max_{\{a_{k,t}\}} \sum_{k,t} u_k(a_{k,t}) \quad \text{s.t.} \quad \sum_{k,t} c_r(a_{k,t}) \leq B_r \ (\forall r), \quad S(\{a_{k,t}\}) \leq \varepsilon. \tag{7}$$

A Lagrangian form adds a risk multiplier $\mu \geq 0$:

$$u_k(a) \ \mapsto \ u_k(a) \ - \ \mu \, s(a), \tag{8}$$

where $s(a)$ is the per-action risk contribution (e.g., skipping verification, calling external tools, writing memory). This makes "trust" compatible with the same allocator logic: verification/trust checks become privileged actions whose use is optimized subject to explicit risk budgets.

**Single-application case**   With $K = 1$, co-optimization still applies because the decision is *intra-application*: FMOS allocates resources across the application's own components (retrieve vs. verify vs. generate vs. tool-use), and across concurrent sessions/agents, under shared budgets and risk constraints.

### A.4. 3) Reusability across enterprises (interface contract + approximate equivalence)

**VFM ABI: program to the interface, not the implementation.**   A core virtualization promise is that applications target a stable interface while the substrate may change. For FMOS, applications program to a VFM "ABI" (API + semantics) independent of the underlying physical FMs, vector stores, tools, or verifiers.

Let $S_P$ be the physical FMOS state (models, caches, indexes, policies, tool handles) and $S_V$ the virtual state exposed to applications (virtual memory/context, virtual budgets, virtual trust guarantees). FMOS implements a mapping $f : S_P \to S_V$ such that for any application-visible operation sequence $e$ there exists an FMOS-internal realization $e'$ satisfying an interface-commutation condition:

$$\boxed{f\big(e(S)\big) \ \approx_{\mathcal{C}} \ e'\big(f(S)\big)} \tag{9}$$

where $\approx_{\mathcal{C}}$ denotes *approximate equivalence under a contract* $\mathcal{C}$ (e.g., budgets respected, provenance attached, safety policy enforced, memory consistency semantics, and task-level acceptance metrics). We use $\approx$ (not strict equality) to acknowledge stochastic generation and changing model backends.

**Why this yields enterprise reuse.**   Eq. (9) formalizes that applications depend on VFM semantics, not physical realization. Therefore, *to the extent that FMOS maintains the contract $\mathcal{C}$*: (i) agent code ports across organizations with different model stacks, (ii) domain capabilities can be packaged as FMOS "drivers" (retrievers, memory schemas, verifiers, policy modules), and (iii) upgrades to physical models/tools can occur with limited application changes—provided the VFM contract remains stable and sensitive operations continue to trap via (1).

### A.5. Summary

- **Co-evolution:** FMOS centralizes privileged mechanisms and learns them from pooled traces; generalization improves with total trapped samples $N$ (Eq. 3). This holds across many apps ($K > 1$) and over time within one app ($K = 1$).

- **Co-optimization:** FMOS acts as a global allocator for coupled budgets and risk constraints; shadow prices coordinate local choices into system-level policies under standard assumptions (Eqs. 4–7).

- **Reusability:** FMOS provides a stable VFM contract; approximate interface-level equivalence enables portability and upgradability without claiming identical outputs (Eq. 9).

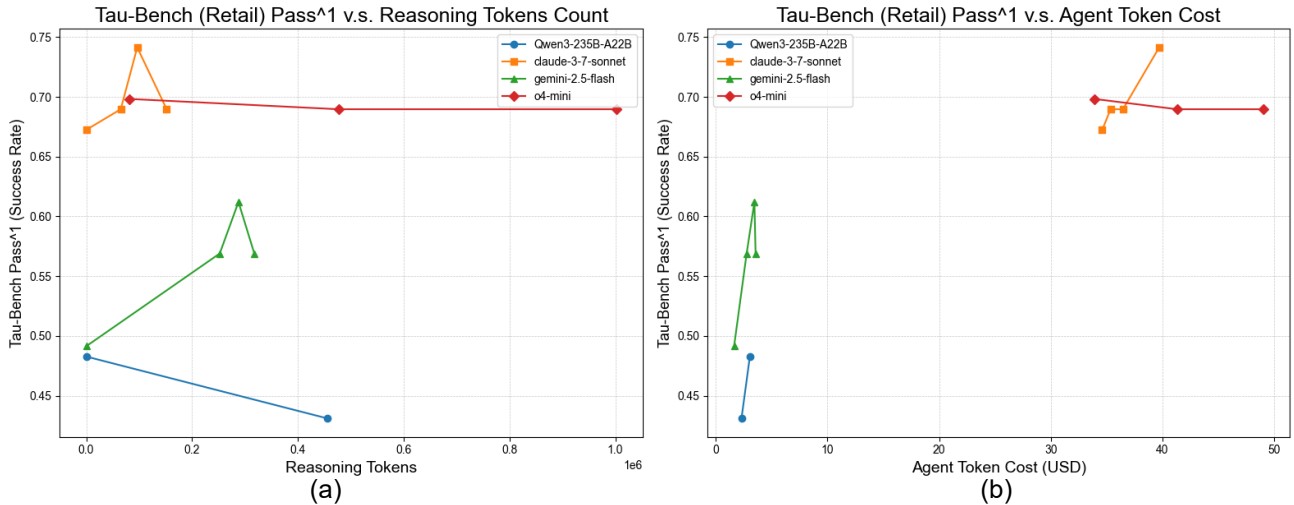

*Figure 4.* Tau-bench (Retail) success rate (Pass^1) vs.: (a) reasoning tokens in agent execution and (b) total cost of all tokens consumed by agent LLM. It can be seen that increasing the number of reasoning tokens (and budget) does not guarantee improved performance.

## B. A Control Reasoning Knob to Optimize Agent Execution

Figure 4 contrasts the *success rate* (Pass^1) of $\tau$-bench (Yao et al., 2024) with (a) the *reasoning tokens* produced by an agent LLM and (b) its total *input/output token cost*. The study spans four recent reasoning LLMs—`Qwen3-235B-A22B`, `gemini-2.5-flash-preview-05-20`, `claude-3.7-sonnet`, `o4-mini`—each exercised under multiple "reasoning–budget" settings. Two clear trends emerge:

1. **Moderate deliberation improves reliability.** Most LLMs except `Qwen3-235B-A22B` exhibit a lift when moving from *no-thinking* to a modest level of reasoning tokens: e.g., `gemini-2.5-flash` rises from 0.49 (no reasoning budget) to 0.61 Pass^1 with a 4096-token budget.
2. **Excessive reasoning degrades performance and cost efficiency.** Beyond a task-dependent "sweet spot" the accuracy curve turns downward while cost grows super-linearly. For example, increasing the `gemini-2.5-flash` reasoning budget to 16,384 tokens actually lowers Pass^1 to 0.57, even though token usage grows by more than four times. Similarly, `o4-mini` reaches its limit at the `low` effort setting; switching to `high` raises the cost per run by 50% without improving performance. `Qwen3-235B-A22B` does not offer reasoning-level control, and enabling reasoning mode causes the LLM to overthink, dropping Pass^1 from 48.3% to 43.1%.

The challenges of setting up these experiments highlighted the heterogeneity of control knobs across models: A practitioner cannot simply "dial in" the same budget across different models. For example, `gemini-2.5-flash` and `claude-3.7-sonnet` allow the setting of an explicit reasoning token budget, while `o4-mini` offers only three opaque effort levels (`low|medium|high`) without a direct token cap. Some open-source LLMs, such as `Qwen3-235B-A22B`, provide only a binary `reasoning on/off` switch. Fine-grained control might be achieved with techniques like Chain-of-Draft (Xu et al., 2025b), latent-space reasoning (Hao et al., 2024), appending tokens such as "Wait" to extend reasoning or using end-of-thinking delimiters like "Final Answer:" to shorten it (Muennighoff et al., 2025; Aggarwal & Welleck, 2025). However, integrating these techniques into a specific LLM serving framework can be non-trivial for application developers. This heterogeneity leaves today's agent developers with brittle, model-specific heuristics that must be hand-tuned for every workflow and re-tuned as models evolve. Worse, mis-configuration can induce over-thinking (Li et al., 2025b), wasting compute while reducing correctness.

These findings motivate the *Learn-to-Reason* in FMOS's Trust & Reasoning hierarchy (Section 4). By abstracting the notion of reasoning effort behind a `set_reasoning_level()` API, FMOS can: (a) **normalize heterogeneous control** that maps the user's high-level budget request to the appropriate switch (`on/off`, effort level, or token cap) for each concrete LLM; (b) **self-evolve budgets on-line** by monitoring success signals, execution trace, and cost to iteratively converge to the task-specific sweet spot. In short, *reasoning is a double-edged sword*: indispensable for difficult tasks yet detrimental when uncontrolled. An OS-like abstraction layer that virtualizes reasoning budgets, just as classical OSs virtualizes memory,

enables cost-aware and model-agnostic optimization of agent performance.

## C. Virtual System Environment Services Enabling Agent Workflows

A virtual FM system must support a prototypical user workflow (see Figure 2) comprising input context preparation, model execution passes, and output processing in a manner that allows for system-level interception and control. Such interception enables a suite of environment services that abstract and manage these stages, for simplifying the development of FM-based agent applications and for enabling the system's capacity for self-evolution.

### C.1. Agent Filesystems

Recent work on *agent filesystems* proposes such an abstraction by building an OS-like filesystem substrate tailored for AI agents (Enberg & Costa, 2025; AgentFS, 2026a). Instead of scattering agent state across ad hoc databases, logs, and local files, these systems encapsulate an agent's runtime artifacts, key–value state, and tool-call audit trails behind a familiar filesystem interface, enabling post-hoc inspection, debugging, and reproducibility (Enberg & Costa, 2025).

AgentFS, for example, implements an agent-oriented filesystem on top of a single SQLite file, making an agent session portable and snapshot-friendly while supporting queryable audit logs for observability and compliance (Enberg & Costa, 2025; AgentFS, 2026b). Isolation mechanisms such as copy-on-write overlays allow agents to safely use real command-line tools without mutating the underlying host project until changes are reviewed and applied (AgentFS, 2026b). This "single durable artifact" design also makes it practical to fork state for subagents and to time-travel (rollback) during development and evaluation (AgentFS, 2026b; Enberg & Costa, 2025).

Nexus generalizes this direction into a programmable, backend-agnostic filesystem for AI agents that combines file storage, memory across sessions, fine-grained (relationship-based) permissions, and semantic search under a unified API (Nexi, b;a). This consolidates several system concerns—persistent memory, access control, and multi-agent sharing—into one substrate that can be deployed locally or in multi-tenant settings (Nexi, b).

From the perspective of context engineering, filesystems provide agents with an interface to store, retrieve, and update an effectively unbounded amount of context without bloating the prompt window (Huang, 2025). Deep agents can offload large tool outputs (e.g., web-search dumps) to files and selectively pull back only the needed spans using filesystem search primitives such as `ls`, `glob`, and `grep` (Huang, 2025). Files also serve as a natural mechanism for long-horizon plans, subagent handoffs, and skill/instruction libraries that can be loaded on demand rather than permanently occupying the system prompt (Huang, 2025). Finally, because agents can write to their own filesystem, user feedback and operational lessons can be persisted as editable artifacts, providing a concrete substrate for longitudinal self-improvement via versioning and rollback (Huang, 2025; Enberg & Costa, 2025).

Within an FMOS architecture, agent filesystems complement the Data Agent by providing a durable memory tier and audit substrate shared across workflows and agents. They operationalize system-level policies (e.g., size-based offloading, skill paging, and trace capture) while supplying OS-like primitives—permissions, versioning, and event triggers—that are essential for safe, governable, self-evolving VFMs (Nexi, b; AgentFS, 2026b).

### C.2. Context Management and Knowledge Augmentation

A FM combines internal (parametric) knowledge acquired during training with (non-parametric) knowledge that it receives as input context (prompts). System environment services control this context both to elicit (selectively focus on) what the model knows and to expand (augment) it with external knowledge.

**Context memory management** methods (Packer et al., 2023; Mei et al., 2024; Asai et al., 2023) "virtualize" limited LLM context window space by retrieving or swapping appropriate content in/out from a persistent store, enabling the creation of stateful agents. Letta/MemGPT (Packer et al., 2023) achieves this using an elaborate system prompt that "teaches" an LLM to summarize, recall, and edit information in its context memory using a toolbox of functions. AIOS (Mei et al., 2024) splits conversation context into blocks and use a k-LRU policy. Self-RAG (Asai et al., 2023) fine-tunes the target LLM with retrieval and critic tokens to trigger retrieval and assess value of retrieved information. These techniques remain useful even for FMs supporting long contexts by filtering irrelevant data and saving costs. However a key challenge in realizing context memory management services lies in automatically learning to identify and focus on what is relevant, which may vary across different domains and use cases.

**Missing interface for mapping context heuristics into the *context management policies*:** This missing interface matters because application developers often *know* which pieces of context are valuable (and when), but cannot express that knowledge to the serving layer that owns the prompt budget. Concretely, developers may want to specify rules such as:

1. **Web research agent (post-answer offload):** After a scraped page has been fully used to answer the local question, proactively offload the full page content to a file and keep only a pointer plus a brief summary in the active window. Today this is typically implemented manually at the application level; however, since the harness already performs compaction/offloading, the same rule could be enforced at the serving layer and reuse any underlying memory tier (files, vector stores, databases) uniformly.

2. **Enterprise infrastructure agent (size-based tool-output offload):** If an MCP server returns a JSON result larger than 20KB (developer-specified threshold), store the payload in a file and insert only a pointer + schema/summary into the prompt before the next action. Current frameworks do not offer a portable way to bind such application-specific thresholds to the underlying context manager.

3. **Adaptive Agent skills unloading:** After a skill has been used, unload the corresponding `SKILL.md` from active context (even if it might be needed later), retaining only a compact "capability header" and a retrieval handle. This becomes important as skills evolve and become lengthy; yet serving frameworks have ad-hoc and non-programmable defaults for progressive skill loading.

4. **Deep research agents (retain thoughts, prune observations):** Empirically, aggressively pruning accumulated web-search/tool results while preserving the agent's internal reasoning trace can improve final outputs; MiroThinker operationalizes a related principle via recency-based retention of tool responses while preserving the full thought/action trajectory (MiroMind Team, 2025). Today, such policies are mostly hand-engineered in application code instead of being declaratively enforced at the serving layer where they could be reused across tasks.

**Knowledge compression and retrieval:** Typically, indexing and retrieval of external knowledge, prior to context augmentation, is achieved using multiple components such as embedding models, retriever models and ranking models, sometimes jointly tuned along with the target FM. Yang et al. (Yang et al., 2024) fine tune a base LLM to compress and retrieve augmented knowledge in terms of a hierarchical state representation, for lifelong context management across sessions, while MemTree (Rezazadeh et al., 2024) maintains a dynamic tree structured memory representation. System environment services can optimize encoding and retrieval performance and resource efficiency, based on workload patterns and context (e.g. context aware pre-fetching, caching, or switching encoding and retrieval algorithms).

**Knowledge oriented abstractions for different modalities:** External knowledge/data may available in a variety of structures and modalities. Further, many scientific explorations involve large scale data and a continual influx of new observations and data. System environment services can enable the efficient curation of such data into knowledge oriented abstractions suitable for augmenting FMs. This curation may in turn be aided by using an FM, and could require optimizations to support high throughput and scale. Different scientific domains and even different scenarios for the same scientific domain may require slightly different abstractions.

### C.3. Reasoning and Trust Augmentation

Reasoning capabilities are essential both in the discovery or evolution (and integration) of new knowledge and when leveraging existing knowledge, tools, simulators etc and for ensuring that FMs generate safe, responsible and trustworthy results. System environment services can control FM output selection and processing (e.g., through constrained decoding, sampling, invoking verification and planning tools, representation engineering (Zou et al., 2023; 2024)), either at the final or intermediate layers.

**Expand and manage reasoning resources (system 1, system 2):** FM augmented reasoning and planning may be characterized into different modes, analogous to human cognition, e.g., a fast thinking *System 1* (e.g., a direct inference) and a slow deliberative (multi-step) thinking *System 2* (Kahneman, 2011). These modes have different resource (and reliability) profiles: System 2 places a heavier load on inference time resources, while System 1 improvements require training and fine tuning resource. System services which control and activate suitable modes of reasoning as neededB, would help enable effective tuning, management and sharing of reasoning resources across applications.

**Reasoning at multiple levels (tiers): Abstract and specialized reasoning:** Answering complex questions and formulating hypotheses in science requires multi-step, hierarchical reasoning that draws from different domain specific concepts. Effective reasoning process templates can be stored and enhanced over time by the system in procedural memory. Analysis of token probability distribution (and associated properties such as entropy and variance) can used to guide reasoning

decisions, for instance by providing insight into how certain tokens dominate or diversify the reasoning space (Besta et al., 2025).

**Low-overhead verification, protection and steering mechanisms:** When FMs produce unreliable, biased or unsafe outputs this could have cascading implications in autonomous FM agent workflows. External verifiers, evaluators, critics and guardrail models/agents are often used to moderate and control FM outputs, along with alignment techniques built into FMs Controls close to the source ensure broad protection but have limited context and higher resource costs. Workflow-specific checks reduce false positives and negatives but are harder to generalize and leave other workloads vulnerable. System level services can be designed to tackle these trade-offs, ensuring consistency, relevance, flexibility and efficiency in verification, protection and steering mechanisms, including measuring and tracking uncertainty of reasoning paths.

## C.4. Model Resource Sharing and Orchestration

FM inference and adaptation demand significant GPU resources, which escalate with reasoning stages, RLMs, and multi-agent setups, causing resource contention. Using smaller FMs, distilled models, and optimized orchestration can mitigate these issues. The underlying model inference/serving platforms typically perform several optimizations for all requests to a given FM, but system environment services can intercept them (Abhyankar et al., 2024) and use its awareness of higher level intent enabling much deeper co-optimizations and management of tradeoffs involved in both model selection and orchestration.

**Scheduling and mapping:** Given a pool for underlying FMs and tools, and a set of application agents/workflows, there are two primary aspects of scheduling to be considered: mapping requests from application agents to one or more suitable FMs from underlying pool of FMs (analogous to mapping process threads to CPU and accelerators) (Ong et al., 2025; Shnitzer et al., 2023)and allocation and scheduling of these FM resources across user agents/FM applications (analogous to scheduling / context switching between user processes and threads)(Mei et al., 2024). System environment services that have more direct context can learn to guide both of these optimizations better and pass them as hints to underlying subsystems such as inference platforms, which in turn can leverage underlying base OS (Prabhu et al., 2025).

**Model composition and instantiation:** In domain specific research where certain models may possess deep domain capabilities that are valuable for a given application but lack important capabilities which are present in other models. System environment services can enable the creation, configuration and provisioning of suitable composite FMs or distilled FMs to meet both capability augmentation and resource/latency constraints (e.g. for agents that perform in the loop steering of experiments or high throughput IT system events monitoring).

**Profiling, measurement, and tracing:** In order to optimize, evolve and control FM systems effectively, new profiling and tracing services would be needed that provide visibility into key FM states and operational conditions.

## C.5. Broader Considerations

Beyond application and system considerations related to the three elements and their interactions at any given point, the design of an operating system for FM workloads in open ended domains also involves some broad long term considerations.

**Continual self-evolution:** The world is always changing. The ability to evolve and adapt with these changes is especially important for system environments that support FMs in scientific discovery and other open ended domains, where new observations, new scenarios to learn from keep emerging and new knowledge is being constantly being generated, verified and refined.

**Continual adoption of latest techniques:** The world of AI also continues to advance at a phenomenal pace (calling into question the shelf life of a position paper like this). Systems environments that support emerging FM workloads, therefore need to be designed with this reality in mind, and must be able to continually adopt better models, frameworks and methods at the same pace, so that every workflow automatically benefits from those advancements.

**External control:** System environments should have mechanisms for external controls to be applied automatically across all workflows by administrators to allow incorporation of global policies, particularly with respect to safety guidance, compliance and resource bounds, e.g. using techniques for incorporation of privileged instruction hierarchy in FMs (Wallace et al., 2024)

# D. Additional Alternative Views

*AI workloads are not that different; few if any changes are needed to conventional OS concepts and methods.*

The FMOS is built as a layer over a conventional OS (Mei et al., 2024); hence it can use insights from observing these resources to provide hints to the underlying OS, as well as control the agent application environments using OS-inspired principles (Mei et al., 2024; Packer et al., 2023). This opens up fresh approaches to address a classical cross-layer dichotomy in OS design: how to provide workload intent to the OS, and system resource awareness to workloads, without breaking abstraction boundaries. Bridging this gap enables better coordination between workflows and the system in order to make the best decisions at both the application and system level.

*Model Context Protocol (MCP) and Agent to Agent communication Protocol advancements will address most of the challenges*

The community has progressed from directly interacting with Foundation Models and devising ways to establish context (such as RAG, GraphRAG etc.) to perform tool calling (introduced by OpenAI in 2023). This led to a cacophony of orchestration frameworks (Langchain, Langgraph, Langflow, n8n, etc.) each on a journey to create a viable ecosystem of libraries/components/modules to enable users and developers to select between different LLM providers (cloud or on-prem), libraries to interact with product or service/providers. This led to tools, resources and prompts being created and a flourishing ecosystem evolved in except that these implementations were captive to their orchestration framework. The introduction of Model Context Protocol as an unifying standard to connect various Agents to business product/service tools, exposing prompts and resources has been a recent game-changer.

An AI agent (typically acting as an MCP client) can now interrogate MCP Server(s) over JSON-RPC to list and execute tools (actionable functions), list and express prompts (interactive templates) and list and provide resources (data). This has enabled an explosive growth of MCP servers of various persuasions across the industry. A parallel contribution has been the introduction of the Agent-to-Agent (A2A) protocol which enables Agents to advertise their capabilities through a template endpoint (./well-known/agent.json). The combination of these two enables an agent to subscribe to one or more MCP servers (Figures 5 and 6); and one/more such agents being able to interoperate in a standardized fashion with each other.

These developments have enabled agents and multi-agent systems in the personal assistant space to become wildly successful. However, these still does not address all requirements required for reliability and scalability in Enterprise computing environments. We outline a few such requirements and therefore justify why the VFMOS approach outlined in this paper might be a more suitable option.

1. **AuthZ/AuthN:** The first generation of agents have evolved from developer centric workflows. Several have automated tasks which otherwise needed multiple steps in an IDE and/or access to a Cloud resource. In those circumstances directly passing API Keys (with developer equivalent credentials) or OAuth2.1 credentials with callbacks (good enough for a human to access) was sufficient and passed on to an assistant. However this causes problems for more sophisticated enterprise products or services where tighter Authentication and Authorization separation is necessary for a downstream tool. Recently Authorization support with OAuth 2.0 has been adapted into the MCP spec for HTTP transport (SSE Server Side Events) (Model Context Protocol Team, 2025).

2. **Agentic Guardrails:** Enterprise applications typically require Service Level Agreements (SLAs) and are tightly coupled iwth one another. Exposing an AI agent to autonomously make changes into these systems exposes the environment to excessive risk. There is a need to intercept the agent-tool, agent-data, and agent-LLM interactions and add appropriate guardrails in a similar fashion as with LLMs. This might differ based on the class, degree of sophistication and need of the downstream Enterprise application or service. This is necessary to develop since it is easy to cause irreversible changes to Enterprise platforms and databases with a malformed prompt, hallucinating LLM or a badly implemented tool function exposed by an MCP server. Finally, since there is no established marketplace for MCP servers, there are often tens of MCP servers for the same application or cloud-service in open-source (and progressively getting worse). This requires a rethink to VFMOS structured approach as advocated earlier in this paper.

3. **Observability, Monitoring and Audits:** It is necessary to debug, trace and monitor, agents and agentic workflows during their entire lifecycle (design and operation). An entirely new set of observability tools such as LangSmith and Langfuse are now being created in the application layer (where modern agent development is taking place). While they are necessary, they will lead to "software bloat" at a fundamental level. Bringing agentic AI development into the VFMOS realm as advocated in this paper would therefore be a better approach.

4. **Transparency, Reliability and Trust:** It is well known that the best performing Foundation Models can still hallucinate.

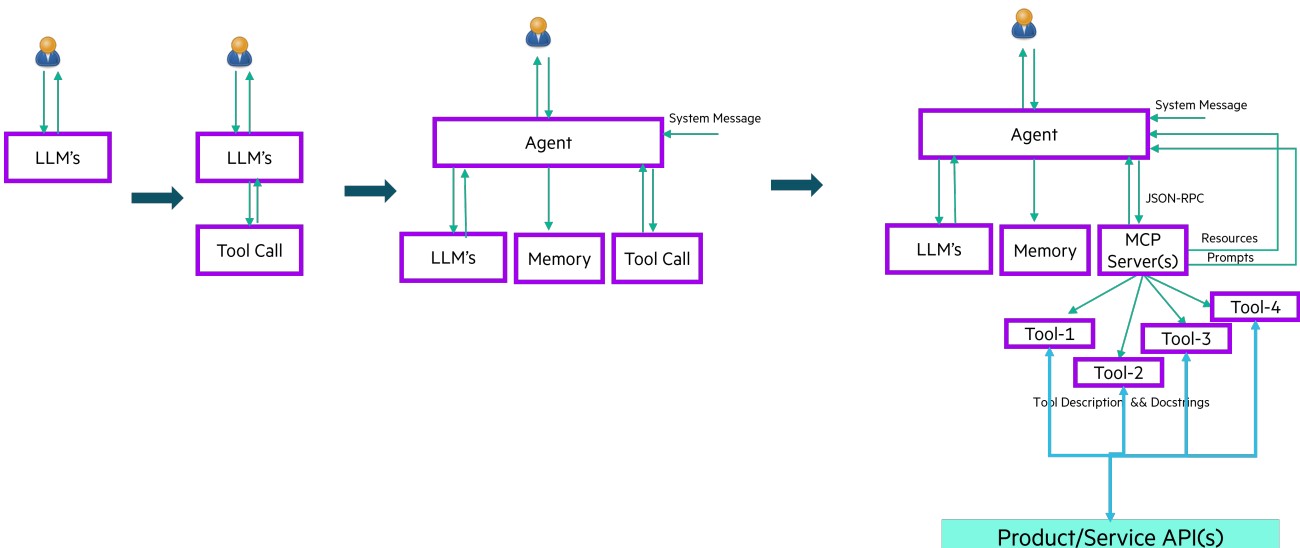

*Figure 5.* **Evolution of Tool calling with LLMs:** Architectural progression from LLMs performing tool-calling to orchestration platforms to the construct exposed by MCP servers

Since FMs are essentially the brains of a modern AI agent, it is therefore inherently untrustworthy. Special attention has to be made at various layers of the hierarchy (such as tool description/Docstrings, system/user prompts, resources exposed via an MCP server) to bring an element of dependability and reliability in agents. While observability, monitoring and auditability are necessary features, it is unlikely to be added into the MCP or A2A standards since they will likely make the spec bloated and unsustainable. This also calls for a rethink in the design of future agentic AI systems along the VFMOS approach and interface advocated in this paper. In this way a dedicated Trust agent can be built and augmented independent of the MCP server or Agent and the appropriate fine-tuning or policy or guardrails implemented.

5. **Performance and Multi-tenancy:** First generation agentic AI workflows have focused on functionality over performance. Much of the performance optimizations have been limited to the AI inference layer. An end-to-end evaluation needs to be performed from the workflow being triggered to the chat prompt being generated. Modern containerized stacks enable inference, MCP servers, underlying service and the agent (MCP client) to be physically and logically separated across network segments. This will be unsustainable to optimize without a standardized VFMOS construct. Finally, most first generation agents are built to run single instances per client. The next generation of agentic workflows will likely need to support multiple tenants in a single instance for which no easy enhancements can be done to prevailing MCP and A2A specifications without bloating them unsustainably.

6. **Client experience:** The overall experience with an MCP-enabled agentic workflow unfortunately depends on which parts of the MCP spec are implemented by it. The MCP Client Feature support matrix (Model Context Protocol, 2025) shows a wide variety of clients with varying levels of integration with MCP servers. This is unsustainable for the ecosystem in the long-term.

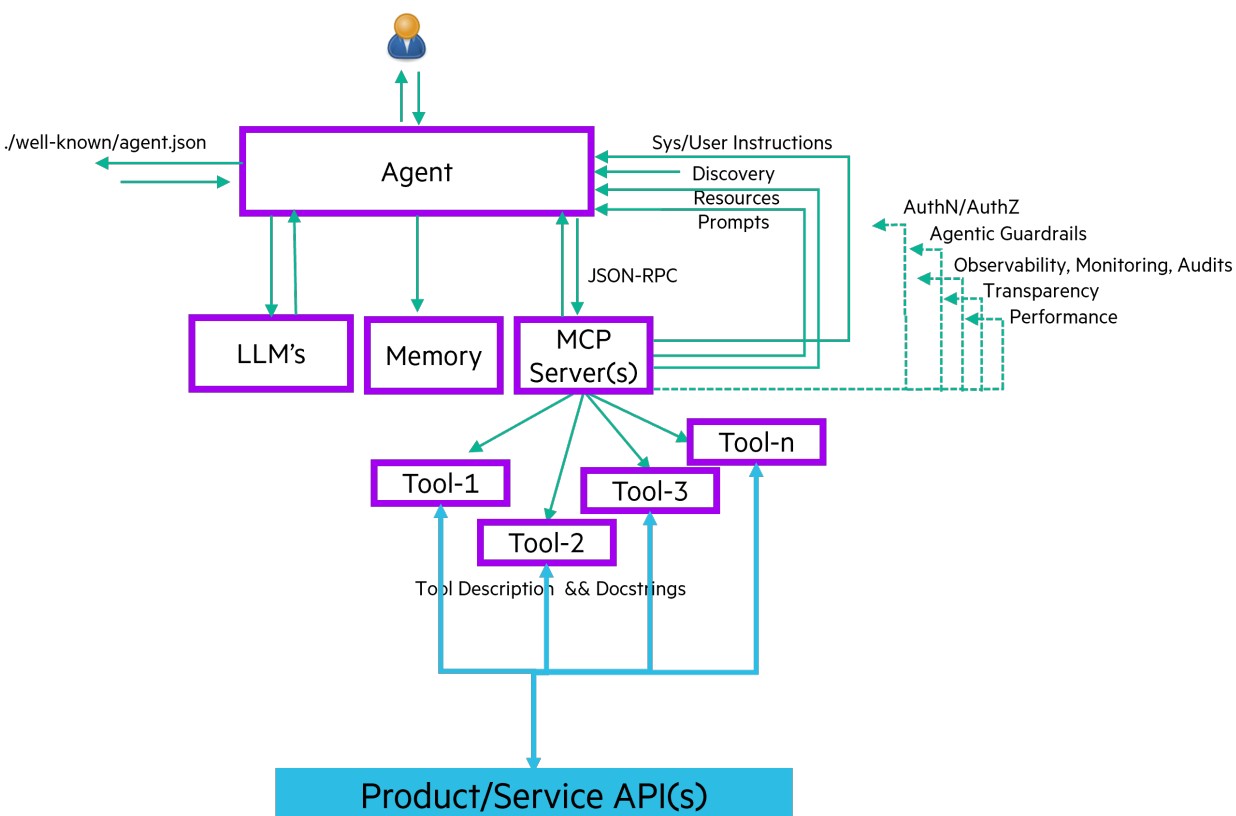

*Figure 6.* **Looking into the future with MCP:** Constructs exposed by MCP protocol today, limitations and future possibilities

