# OpenReview forum: "Position: It is Time to Virtualize Foundation Models with a Self-evolving Operating System Layer"
_ICML.cc/2026/Position_Paper_Track — ICML 2026 Position Paper Track regular_

### Official Review · Reviewer_cdUq · 2026-03-12

**Significance:** 4
**Argument Clarity:** 4
**Rating:** 4
**Confidence:** 4

**Questions:**

1. Self-evolution is described as the key difference of FMOS over classical virtualization, yet the paper provides no formal or empirical treatment of how it works in practice. The shared policy learning argument (Appendix A.2, Eq. 2–3) is a standard multi-task learning bound and does not specifically justify the FMOS architecture over, say, a well-engineered shared library. The bound shows that pooling data improves estimation, which is true of any shared learning system, not uniquely of an OS-like virtualization layer. The authors should articulate what specifically about the virtualization boundary about the claimed co-evolution benefits.

2. The partition of operations into innocuous vs. sensitive (A_ino, A_sen) is critical of the formalism in Appendix A, yet it is left undefined. Can the authors provide a concrete, non-trivial example of this partition for a realistic agent workflow, and explain how the FMOS learns or discovers this partition at deployment time?

3. How does the proposed FMOS differ architecturally from a well-engineered shared middleware layer (e.g., a shared RAG service plus a shared routing proxy) that does not invoke the virtualization abstraction? What does the virtualization boundary specifically buy that a service-oriented architecture does not?

4. The co-evolution claim (Appendix A.2) relies on the standard multi-task learning sample-complexity argument. Can the authors clarify what is unique to the FMOS framing that makes this argument stronger than it would be for, say, a shared fine-tuning or shared prompt-optimization service?

5. The paper claims FMOS is compatible with existing frameworks via standard API hooks. Could the authors sketch a minimal prototype implementation using an existing framework (e.g., LangGraph + an OpenAI-compatible endpoint) to demonstrate that the trap-based interception is practically realizable without unacceptable latency overhead?

**Alternative Views Section:**

Yes

**Compliance With Llm Reviewing Policy A Conservative:**

Affirmed.

**Discussion Potential:**

3

**Final Justification:**

I keep my positive rating.

**Paper Summary:**

This position paper argues that the transition from single foundation models (FMs) to compound agentic systems has exposed a critical missing abstraction layer in the AI software stack. Therefore, the authors propose a Foundation Model Operating System (FMOS) whose primary abstraction is the Virtual Foundation Model (VFM). A VFM presents applications with the illusion of a dedicated, trustworthy FM instance with effectively unbounded capabilities, while the FMOS internally manages context, memory, model routing, verification, and policy enforcement. The paper further argues that  the FMOS introduces progressive quality gain through self-evolution as a first-class design principle.

**Position:**

Yes

**Position In Title:**

Yes

**Related Work:**

4

**Strengths And Weaknesses:**

The VFM abstraction is a good conceptual framework, this paper provides clear motivations and descriptions with actionable call to actions.

**Support:**

3

---

> ### Author Rebuttal · Authors · 2026-03-31
>
> We thank the reviewer for positive feedback and assessment of the formalism. We respond to each question below:
>
> $\textbf{Q1: On Self-evolution, shared policy learning argument, FMOS vs shared library?}$
>
> We agree that Eq. (3), taken in isolation, is a standard generalization bound, and we do not claim it alone justifies FMOS. The FMOS-specific argument is the combination of Eq. (1) and Eq. (2).
> Eq. (1) imposes the virtualization condition A_sen ⊆ A_priv, i.e., every operation affecting shared budgets, shared knowledge state, or trust state must be mediated by FMOS. Once this holds, Eq. (2) is no longer an arbitrary shared learner; it is a policy learned over a common privileged trapped surface (retrieval escalation, memory updates, verifier invocation, and model routing). A shared library lacks precisely this: it may pool data or reuse code, but it does not impose a mediated control plane over system-critical operations. Without Eq. (1), the learned object remains framework-local or application-local; with Eq. (1), Eq. (2) becomes a system policy over the shared FMOS control surface. Eq. (3) is then used only to explain why learning that centralized policy improves faster from pooled trapped-operation traces.
> In short, the novelty is not a new statistical bound; it is that virtualization makes Eq. (2) the correct shared object to optimize at the system level. We will revise the appendix to make this architectural dependence explicit.
>
> $\textbf{Q2: Undefined innocuous vs. sensitive?}$
>
> Following Appendix A.1, A_ino denotes fast-path operations that do not alter shared FMOS-managed state (e.g., drafting from loaded context, local summarization, formatting). A_sen denotes operations that change or consume shared resources such as shared knowledge state or trust state, and therefore must belong to A_priv and trap under Eq. (1).
> A realistic enterprise-agent workflow makes the distinction clear:
> Innocuous (A_ino) operations include drafting a response from already-loaded context, local summarization, and formatting or paraphrasing an answer already supported by the current context.
> Sensitive (A_sen) operations include increasing retrieval depth, writing to persistent memory, routing to a different FM, invoking an external tool with side effects, or escalating to verification or policy checks.
> These belong to A_sen because each changes or consumes FMOS-managed shared state - such as budget, memory, model allocation, or trust level - and thus must be governed under Eq. (1).
> We agree a concrete example is missing from the paper and will add one to the appendix in the revision.
>
> $\textbf{Q3: FMOS vs a well-engineered shared middleware layer?}$
>
> A shared middleware stack (RAG + proxy) intercepts only specific point services (retrieval/routing), whereas FMOS intercepts all sensitive operations. The key differences are: middleware maintains per-service state with no cross-service consistency, while FMOS maintains versioned, recoverable state across services; trust escalation in middleware is application-level and framework-specific, while in FMOS it is system-level and enforceable; and applications coupled to middleware are tied to service implementation versions, while FMOS provides a stable VFM contract (Eq. 9) under which the substrate evolves freely.
> Stable contract: Applications depending on shared middleware are coupled to service implementation versions (e.g. RAG-service-v2), whereas FMOS provides a stable VFM interface contract (Eq. 9) under which the substrate can evolve freely.
> The core benefit is a stable contract under a changing substrate, improvements propagate without application rewrites, and governance is enforceable rather than advisory.
>
> $\textbf{Q4: On the uniqueness of the co-evolution claim?}$
>
> Eq. (2) is not a prompt-level or weight-level objective - it is a policy over privileged actions (retrieval depth, verifier strength, routing choice, memory tier), governed by the virtualization condition in Eq. (1). Shared fine-tuning or prompt optimization can improve a component, but they don’t centralize sensitive operations under a common mediated control plane. Pooled traces improve the system control policy itself, not just a model or prompt. The stronger claim is architectural: FMOS turns co-evolution from ad hoc component reuse into learning over the system control plane. We will revise the appendix to state this distinction clearly.
>
> $\textbf{Q5: On the minimal prototype?}$
>
> We will include a prototype sketch in the revised version where FMOS is exposed via an OpenAI-compatible Responses endpoint with a thin interceptor. LangGraph (or any framework) routes model calls through an FMOS endpoint with no framework modification. Realizing the trap mechanism admits multiple approaches, for instance, system prompt instructions can leverage the model's own ability to recognize the need of trap, or other mechanisms like representation engineering. Which approach is optimal is an active research direction.

---

> > ### Author Rebuttal · Reviewer_cdUq · 2026-04-02
> >
> > 1. Degradation under violated virtualization. The revised argument makes Eq. (1) load-bearing, i.e.,  the entire co-evolution claim depends on all sensitive operations being mediated. What happens when this condition is partially violated? Does the guarantee degrade gracefully or collapse? A brief discussion of failure modes would strengthen the architectural necessity argument considerably.
> >
> > 2. Static vs. adaptive partition. The A_ino/A_sen examples are helpful, but the partition appears designer-specified and static. In practice, "innocuous" can be context-dependent (e.g., a local summarization that consumes unexpected compute budget arguably becomes sensitive). Is the partition intended to be fixed or adaptive, and if adaptive, what mechanism governs reclassification?

---

### Official Review · Reviewer_qme8 · 2026-03-12

**Significance:** 3
**Argument Clarity:** 2
**Rating:** 4
**Confidence:** 4

**Questions:**

N/A

**Alternative Views Section:**

Yes

**Compliance With Llm Reviewing Policy A Conservative:**

Affirmed.

**Discussion Potential:**

4

**Final Justification:**

My main concerns have been addressed. It would be helpful if the final version of the paper incorporated the points raised in the rebuttal to my review.

**Paper Summary:**

This paper argues that the AI applications require a system layer that virtualizes LLM interactions analogous to how virtual machines abstract physical hardware, giving applications the illusion of dedicated, trustworthy LLM instances with effectively unbounded capabilities. The paper also depicts some desirable features of a virtual LLM environment should have and outlines a potential architecture for an LLM operating system.

**Position:**

Yes

**Position In Title:**

Yes

**Related Work:**

2

**Strengths And Weaknesses:**

Strengths
* While it is hard to define what an LLM OS is, the paper tries their best to depict the desirable features of a virtual LLM environment should have, which is helpful to stimulate constructive, civil discussion.

Weaknesses
* While the paper is strongly connected to systems research, the alternative views section are mainly from the perspective of agent developers and do not reflect the views of the systems community. For example,
  1. Rather than functioning like operating systems, could the system foundations for LLMs eventually play a role closer to that of database systems?
  2. If the system ultimately plays a role similar to an “OS,” it remains unclear whether components such as data agents should be included at the operating system level. Historically, database systems have typically been independent of the operating system and serve multiple applications.
* The related work on model resource sharing and orchestration should also include recent MLSys research that enables LLM systems to support self-evolution through the concurrent execution of fine-tuning and inference [1].

[1] He, Yongjun, et al. "Resource multiplexing in tuning and serving large language models." 2025 USENIX Annual Technical Conference (USENIX ATC 25). 2025.

**Support:**

2

---

> ### Author Rebuttal · Authors · 2026-03-31
>
> We thank the reviewer for the positive assessment of our position and engaging with the systems framing of our work. We address the concerns below:
>
> $\textbf{1) Alternative views don't reflect the systems community. Could the system play a role closer to a database system rather than an OS?}$
>
> This is a valuable point, and we agree it warrants a more substantive treatment in the alternative views section. The database analogy has intellectual precedent — notably Michael Stonebraker's DB-OS work explored precisely the inversion of the conventional relationship, proposing that an OS be built on top of a database rather than the other way around. There are genuine strengths on both sides of that debate, and we will discuss this history more explicitly in the alternative views section to do justice to the systems community perspective.
> That said, we believe the OS framing more precisely captures what FMOS is doing for the following reason. A database system, however capable, is still an external service — an application must explicitly choose to call it, query it, and interpret its results. FMOS, by contrast, is designed to automatically determine when to intervene and when not to, operating transparently beneath the model interface. Crucially, the interface that applications interact with today is a model interface, not a database interface — it involves generation, reasoning, tool invocation, and context management, none of which map naturally onto query semantics. The OS analogy is specifically about creating a virtualization layer that constructs an illusion between what the application sees as the model interface and what is actually executing underneath it.
> On whether data agents should be at the OS level: The historical observation is correct, but we believe the traditional OS itself offers the right analogy for why the data agent belongs in the FMOS layer. Even in classical OS design, there is a principled separation between memory and storage: file systems can be implemented either as part of the kernel or as user-space file systems, and from the kernel it is possible to call out to a database-like component for higher-level storage semantics. Databases, in turn, sit above file systems or block devices and provide additional transactional and query semantics. The layering is not binary — it is a spectrum, and the right placement of each component depends on how tightly coupled it is to execution-time state.
> Context management in FMOS falls firmly on the execution-coupled end of this spectrum. Decisions about what to page into the active context window must be made synchronously with model invocation, and they depend on per-request budget, latency constraints, and trust-state signals that a standalone external database service simply cannot observe. This is directly analogous to why page-replacement policy lives in the kernel's memory manager rather than in an external storage service: the decisions require visibility into live process execution state. The FMOS Data Agent plays the role of the page-table manager; external databases (vector stores, graph DBs) play the role of the disk subsystem beneath it. Importantly, FMOS is layered on top of existing data platforms + data hubs (including data bases etc), inference platforms + model hubs, and API services/tool platforms.
>
> $\textbf{2) Related work on model resource sharing should include recent MLSys work on concurrent fine-tuning and inference}$
>
> We thank the reviewer for this pointer. He et al.'s resource multiplexing work for concurrent fine-tuning and inference is directly relevant to FMOS's FM Composition Optimizer (Section 5.2) and the FM-updater component (continual learning/unlearning). We will add this reference and discuss it in the context of Section 4.5 (Model Resource Sharing) and Appendix C.4.

---

> > ### Author Rebuttal · Reviewer_qme8 · 2026-04-03
> >
> > Thank you for your clarification.

---

### Official Review · Reviewer_ur2W · 2026-03-13

**Significance:** 4
**Argument Clarity:** 2
**Rating:** 5
**Confidence:** 3

**Questions:**

* While the paper advocates about an inherently deployment-time framework, it does affect how models will be used and therefore what needs to feed into the training pipeline for refinement. What would such changes be and how would the current framework enable that?
* Is the existence of an OS-like standardisation expected to change model training? How is it ensured that the most capable models adhere to such interfaces?
* Does the FMOS inherently come with additional latency due to the standardisation of interfaces that might not be compatible with everyone?
* How does FMOS account for user multi-tenancy and resource governance amongst different tools, models and agents?
* How would users be able to effectively debug agent applications under FMOS?

**Alternative Views Section:**

Yes

**Compliance With Llm Reviewing Policy A Conservative:**

Affirmed.

**Discussion Potential:**

3

**Paper Summary:**

This position paper proposed Foundation Model Operating System (FMOS), an abstraction layer to address today's fragmentation in agentic development. Currently, building around compound AI systems relies on diverse orchestration frameworks that replicate functionality and enhance non-portability. To this end, they propose FMOS, a system which exposes virtual foundation models (VFMs) as a dedicated model abstraction with infinite memory, context and capabilities. Their proposed architecture comprises a data agent for context management; a composition optimiser for model routing; and a Trust & Reasoning agent to provide risk-aware, safe execution. Contrary to typical OSs, FMOS also enables self-evolution. Ultimately, the proposed approach seeks to abstract away the dynamics of AI agent development to enable scalability and robustness.

**Position:**

Yes

**Position In Title:**

Yes

**Related Work:**

3

**Strengths And Weaknesses:**

### Strengths

* The paper makes a timely and accurate critique of today's GenAI deployment orchestration frameworks and pinpoints the replication across similar concepts, their brittleness in working across models and use-cases and their non-portability. To this end, they propose FMOS that decouples the application logic from core services.
* The proposal moves past the metaphor and proposes a meaningful partition of concerns to ensure intelligent context and memory management, efficient model routing and safety in execution.
* Unlike traditional systems, the current proposal puts self-evolution as a first-class citizen, allowing system to improve over time without retraining.

### Weaknesses

* While setting the foundations, the paper still suffers from some imprecisions in definitions, such as what a "privileged" or "sensitive" operation is, or how self-evolution works under non-stationary VFMs.
* The standardisation of an abstraction layer to work across many models of varying capabilities, sizes and architectures is a complex undertake that required coordinated efforts and network effects. One size-fit-all might also stand in the way of deploying on different targets, such as on-device, which has very different latency characteristics and use-cases.
* Abstraction and self-evolution will probably make such system increasingly difficult to inspect, debug and replicate.

**Support:**

3

---

> ### Author Rebuttal · Authors · 2026-03-31
>
> We thank the reviewer for the thorough review and for recognizing the timeliness of our work. We respond to each weakness and question below:
>
> $\textbf{W1: Imprecisions in definitions, how self-evolution works under non-stationary VFMs?}$
>
> We will expand these definitions in Section 4.1
> Sensitive operations (A_sen) can change shared system state: e.g., reading/writing memory tiers. Operations like performing in-context arithmetic that affect only local state are innocuous (A_ino).
> Privileged operations (A_priv) are those that FMOS designates as needing interception,  at a minimum, all sensitive ones (A_sen ⊆ A_priv, Eq. 1).
> On self-evolution under non-stationary VFMs: For a new model version, FMOS's versioning (Section 3)  tests policies like updated routing against the new backend before rollout, maintaining the VFM contract (Eq. 9).
>
> $\textbf{W2: Standardization across models of varying capabilities?}$
>
> If reference is to FMOS API, this is compatible with industry-wide standards like OpenAI-API (Open Responses API) endpoints. The abstraction layer does not impose a novel interface on consumers. If reference is to whether FMOS imposes a single model for consumers, we clarify that we do not envision this as "one size fits all." As outlined in the FM Composition Optimizer (Sec 5.2), FMOS materializes a VFM for a consumer scenario/use-case (similar to configuring VMs for specified tenants/use-cases). On-device deployment with different latency profiles would be achieved through VFM materialization strategies (e.g., distilled models with fast-path execution, as in VFM C in Fig. 1).
>
> $\textbf{Q1: changes to the model training pipeline?}$
>
> The FM Composition Optimizer enables model training only when necessary, based on quality or policy requirements. A VFM acts like any other LLM in a consumer pipeline. If agentic requests necessitate tuning based on operational constraints, the tuning/routing implementation is handled internally by the VFM and is transparent to the consumer pipeline, which remains undisrupted.
>
> $\textbf{Q2: OS-like standardization expected to change model training?}$
>
> FMOS enforces its contract at the virtualization boundary. Physical FMs remain provider-specific internally; thin adapters expose a common interface for requirements like messages, or checkpoint/resumption. As an example, constrained decoding and structured outputs enforce syntactic compliance. The core guarantee is that privileged effects are mediated by FMOS.
> FMOS does not assume all capable models are born compliant; it provides compatibility and enforcement while creating incentives for future models to become progressively FMOS-native. We will sharpen this distinction in the revision.
>
> $\textbf{Q3: Additional latency due to interface?}$
>
> The VFM interface is designed to be thin like OpenAI-compatible / Open Responses endpoints, where the abstraction layer doesn’t materially alter the execution path. The FMOS architecture (Sec 5.4) separates a fast path from a slow path. Trap activation on the slow path introduces additional latency only when quality or budget conditions require it, which is a justified trade-off for improved output quality. Ongoing research will investigate the minimization of this overhead through methods like latency-aware scheduling.
>
> $\textbf{Q4: multi-tenancy and resource governance?}$
>
> FMOS addresses multi-tenancy at the Orchestrator. For each request, FMOS constructs a per-request parcel with accumulated metadata in a scoped execution chain. It performs concurrent, asynchronous request handling and resource management realized through a multi-level scheduling technique. Request tasks are categorized into rapid-response (fast path) and optimized-response (trap-triggered slower paths). We acknowledge this to be an important requirement at scale and are consciously devising with a focus on resource optimization.
>
> $\textbf{Q5 + W3: On debugging agents under FMOS?}$
> Unlike unconstrained self-evolution, our position is that FMOS manages self-evolution consistently and makes agent applications easier to debug by presenting a standardized, inspectable system at the VFM boundary.
> Since FM interactions pass through the VFM API, FMOS can emit a causal trace per invocation covering flow path functions like trap activations, enabling layered attribution:  when an agent produces a wrong answer, developers can localize the fault, e.g., in application logic, or in functions like context management.
> FMOS supports checkpointing and replay: the execution state at trap points (e.g., model version) can be recorded for controlled replay. Developers can set breakpoints on semantic events (e.g., before a privileged action) and perform debugging by replaying from a checkpoint.
>
> In the revised version, we will cover this explicitly with a paragraph on debuggability that includes relevant metrics.

---

> > ### Author Rebuttal · Reviewer_ur2W · 2026-04-02
> >
> > Thank you for the detailed rebuttal and clarifications. I have a few follow-ups:
> >
> > i) While I understand the **OAI-compatibility layer**, I believe the challenge extends beyond API-level compatibility, as real-world usage involves:
> > - prompting formats (chat templates, system/user roles, tool calling schemas)
> > - model capabilities (reasoning control, structured output, multimodality)
> >
> > It would be valuable to clarify the compatibility layer beyond the API that normalises the above behaviour and how FMOS avoids collapse on the lowest common denominator capabilities.
> >
> > ii) **Wrt training**, what are the new signals/datasets required for training (traces, routing decisions, verification outcomes)? Does this imply a shift towards system-level training objectives rather than purely model-level objectives? As it stands, the rebuttal suggests transparency to the consumer pipeline, but the manuscript implies non-trivial upstream changes to how models are trained and evaluated. Making this explicit would strengthen the contribution.
> >
> > iii) **On observability**, I appreciate the comment, but system behaviour may become harder to reason about deterministically under learnable components. It would be valuable to pinpoint the desired components that would enable that at scale (even as future research directions).

---

### Official Review · Reviewer_K1Ps · 2026-03-16

**Significance:** 3
**Argument Clarity:** 4
**Rating:** 5
**Confidence:** 2

**Questions:**

How could VFM provide additional "system-level guarantee"? Is it more of an observation of the current status of existing libraries? Or is there a theoretical limitation of libraries versus "OS"?

**Alternative Views Section:**

Yes

**Compliance With Llm Reviewing Policy A Conservative:**

Affirmed.

**Discussion Potential:**

4

**Final Justification:**

I think FMOS is an interesting and important topic to discuss

**Paper Summary:**

This paper argues that we need an additional layer between LLM agentic frameworks, such as AutoGen, and the inference API. This layer should provide "system-layer" guarantees on managing the data, memory, resources, logging, routing, trust, and so on. This layer is named as virtual FM (Foundation Models), and the paper also proposes a specific architecture for it, FMOS.

**Position:**

Yes

**Position In Title:**

Yes

**Related Work:**

2

**Strengths And Weaknesses:**

## Strengths
This paper discusses an interesting and important topic: reliable and governable executions for LLM agents. It discusses this FMOS from many perspectives and provides many alternative views with corresponding responses. It is likely to inspire discussions.

## Weakness
* I'm not sure if LLMs or FMs are stable enough to discuss their OS.
* I'm not an expert in OS or SE, and I cannot tell the difference between VFM and a library. How could VFM provide additional "system-level guarantee"?

**Support:**

3

---

> ### Author Rebuttal · Authors · 2026-03-31
>
> We thank the reviewer for the positive assessment of our work's importance and for raising questions that go to the core of our position. We address each concern below:
>
> $\textbf{1) If LLMs/FMs are stable enough to discuss their OS}$
>
> We thank the reviewer for highlighting this concern, which is actually an argument for the position we are taking: a constantly evolving FM landscape is precisely why a unified, stable interface layer between agents and models is needed now. Applications today are forced to absorb every model upgrade, deprecation, and capability shift individually, with no shared boundary protecting them from churn. FMOS addresses this directly: the VFM interface is designed to be the stable contract that sits between applications and the evolving model substrate, absorbing that evolution so applications do not have to.
> This is also why self-evolution is a first-class design principle in FMOS rather than an afterthought. We are not assuming the landscape will stabilize; we are assuming it will keep evolving, that models will continue to improve and change, and that applications built on top of them will need to evolve in turn. FMOS is the component that enables and mediates that co-evolution: it maintains a stable interface upward toward applications while continuously adapting its internal policies (routing, prompting, verification, memory management) downward toward an evolving model substrate.
>
> $\textbf{2) The difference between VFM and a library. How could VFM provide additional ``system-level guarantee"?}$
> $\textbf{Is it more of an observation or is there a theoretical limitation of libraries versus `OS'?}$
>
> The distinction between VFM and a library goes beyond functionality and lies in how learning, control, and guarantees are realized across agents and applications. A key differentiator is that libraries operate within the scope of a single application or agent. Any improvements - whether in prompting, retrieval strategies, routing heuristics, or safety policies - remain local unless explicitly exported, versioned, and redeployed. As a result, learning is fragmented and does not naturally accumulate across workloads.
> In contrast, FMOS introduces a shared system layer with a unified interception boundary (via VFM endpoints). This enables the system to: (a) continuously collect execution traces across all agents and applications; (b) learn policies (e.g., when to retrieve, how to route models, when to escalate for verification) from this aggregated experience; and (c) propagate these improvements system-wide without requiring application-level changes.
> This is what enables self-evolution both within an agent (over time) and across agents (via shared learning). From a systems perspective, this capability arises because FMOS enforces a centralized control plane through trap-based interception, rather than relying on voluntary library calls. As formalized in Appendix A, this aligns with classical virtualization principles: operations that affect shared resources or system behavior must pass through a common mediation layer. This is what allows FMOS to both enforce system-level guarantees (e.g., budgets, trust policies, memory consistency) and learn from global interaction data.

---

> > ### Author Rebuttal · Reviewer_K1Ps · 2026-04-04
> >
> > Thanks for the clarification. I think FMOS is an interesting and important topic and would like to raise my score to 5 accordingly. I'm not sure if LLM is good enough for self-evolution now, though.

---

### Decision · Program_Chairs · 2026-04-30

**Decision:**

Accept (regular)

**Comment:**

This paper presents a timely, well-argued position that the AI community must move beyond fragmented agentic frameworks toward a Foundation Model Operating System utilizing a Virtual Foundation Model abstraction. The reviewers appreciate the relevance of the problem and the proposed OS analogy. During the rebuttal, the authors successfully addressed all major reviewer concerns, including clarifying the technical distinctions between their virtualization approach and standard shared libraries, expanding on the database vs. OS alternative perspective, and refining their formalisms regarding sensitive versus innocuous operations. Given its quality and strong potential to inspire future systems research in agentic AI, an accept is recommended.